# NExCO: Native Solution Expansion for Diffusion-based Combinatorial Optimization

**Yu Wang[1], Yang Li[2], Jiale Ma[2], Junchi Yan[2*], Yi Chang[1,3,4*]**

[1]School of Artificial Intelligence, Jilin University
[2]School of Artificial Intelligence, Shanghai Jiao Tong University
[3]International Center of Future Science, Jilin University
[4]Engineering Research Center of Knowledge-Driven Human-Machine Intelligence, MOE, China

## Abstract

One central challenge in Neural Combinatorial Optimization (NCO) is handling hard constraints efficiently. Beyond the two classic paradigms, i.e., Local Construction (LC), which sequentially builds feasible solutions but scales poorly, and Global Prediction (GP), which produces one-shot heatmaps yet struggles with constraint conflicts, the recently proposed Adaptive Expansion (AE) shares the advantages of both by progressively growing partial solutions with instance-wise global awareness. However, existing realizations bolt AE onto external GP predictors, so their solution quality is bounded by the backbone and their inference cost scales with repeated global calls. In this paper, we fundamentally rethink adaptive expansion and make it native to a generative model, acting as its intrinsic decoding principle rather than an external wrapper. We propose NEXCO, a CO-specific masked diffusion framework that turns adaptive expansion into the model's own iterative unmasking process. Specifically, it involves a solution-expansion training procedure with a time-agnostic GNN denoiser, which learns diffusion trajectories between fully masked solutions and ground-truth solutions. With the trained time-agnostic denoiser, we introduce a novel solution expansion scheme at the solving stage, enabling adaptive control over the intermediate solution states. It is achieved by constructing candidate sets according to confidence scores and applying feasibility projection to expand the solution while respecting constraints. In this way, "adaptive" is not an afterthought but the decoding itself: intermediate diffusion states are meaningful partial solutions and progress is instance-adaptive rather than schedule-bound. Extensive experiments on representative CO problems show that NEXCO achieves approximately 50% improvement in solution quality and up to $4\times$ faster inference compared to prior state-of-the-art solvers. The source code is publicly available at https://github.com/yuuuuwang/NExCO.

# 1 Introduction

Combinatorial optimization (CO) is a sub-filed of mathematical optimization that involves finding the optimal solution from the discrete feasible sets. Due to their inherent NP-hardness, solving large-scale instances efficiently remains a longstanding challenge. Recent progress in Neural Combinatorial Optimization (NCO) has reduced reliance on handcrafted heuristics by learning data-driven solvers (Bengio et al., 2021; Qiu et al., 2022; Sun & Yang, 2023; Li et al., 2024; Ma et al., 2025).

Existing neural constructive solvers fall into two extreme paradigms: (1) *Local Construction (LC)* methods sequentially generates solutions in an autoregressive manner, ensuring feasibility but suffering from myopic decisions and poor scalability (Kool et al., 2019; Kwon et al., 2020; Berto et al., 2023; Drakulic et al., 2023; Pan et al., 2025); (2) *Global Prediction (GP)* methods predict full probability heatmaps in one shot, capturing global structure efficiently but producing smooth distributions that cause noisy decoding and constraint violations (Joshi et al., 2019; Fu et al., 2021; Qiu et al.,

---

*Corresponding authors: Junchi Yan and Yi Chang (yanjunchi@sjtu.edu.cn; yichang@jlu.edu.cn). This work was partly supported by MOST of China (2023YFF0905400), NSFC (92370201, U2341229, 625B2119).

Table 1: **Comparison of NCO paradigms.** Local Construction (LC) and Global Prediction (GP) represent two extremes; COExpander realizes Adaptive Expansion (AE) as a costly wrapper around GP, adjusting decision granularity externally; NExCO makes AE *native*, embedding adaptive expansion directly into a CO-specific masked diffusion for efficient, feasible solution construction. NFEs: number of function evaluations; $T_s$: diffusion inference steps; $D_s$: AE expansion steps.

| | LC | GP | AE | NExCO (Ours) |
|---|---|---|---|---|
| **Decoding granularity** | One-by-one | All-at-once | Adaptive (wrapper) | Adaptive (native) |
| **Partial feasibility** | ✓ | ✗ | ✓ | ✓ |
| **Global awareness** | ✗ | ✓ | ✓ | ✓ |
| **Complexity (NFEs)** | $O(n)$ | $O(T_s)$ | $O(D_s * T_s)$ | $O(T_s)$ |

2022; Min et al., 2023; Sun & Yang, 2023; Xia et al., 2024; Li et al., 2024; Xiao et al., 2024). To bridge LC and GP, the Adaptive Expansion (AE) paradigm was introduced (Ma et al., 2025), which adaptively determines the number of variables per step. While effective, current implementations (e.g., COExpander) are merely wrappers around GP backbones such as Fast T2T, leaving two fundamental issues: (i) performance is bounded by the backbone predictor, and (ii) inference cost scales as $O(D_s \cdot C_{GP})$, far higher than vanilla GP solvers. Table 1 provides a structured summary of these trade-offs, contrasting LC, GP, and AE with our proposed method across multiple dimensions.

This raises a natural question: **can adaptive expansion be made native, i.e., encoded as the intrinsic decoding principle of a generative model, rather than an external wrapper?** Here, native AE would mean: (i) expansion progress and step size are instance-adaptive, driven by model confidence and constraints, without relying on fixed timestep schedules or external GP predictors; (ii) intermediate states are valid, constraint-aware partial solutions, where variable commitments are enforced through feasibility projection rather than deferred to post-hoc heuristic heatmap decoding. Diffusion models appear promising: their iterative refinement resembles constructive expansion. However, existing diffusion-based solvers (e.g., DIFUSCO, T2T/Fast T2T (Sun & Yang, 2023; Li et al., 2023b; 2024)) still operate in the GP paradigm: they generate global probability heatmaps and rely on heuristic decoding, without effectively leveraging intermediate states as partial solutions. We argue that two factors hinder this: (i) intermediate states lack semantic meaning as partial solutions, and (ii) fixed timestep schedules rigidly control denoising progress, preventing instant-adaptive progress.

In parallel, the broader generative modeling community has advanced Masked Diffusion Models (MDMs) as a powerful alternative to autoregressive decoding in large language models (Austin et al., 2021; Sahoo et al., 2024; Ou et al., 2025; Zheng et al., 2025; Nie et al., 2025b), which progressively unmask tokens instead of denoising noise. MDMs provide meaningful intermediate states, allow schedule-free training, and support efficient parallel decoding. *These properties closely match the needs of NCO, making masked diffusion a natural foundation for native adaptive expansion.*

Built on this insight, we propose **NExCO** (Native Adaptive Expansion for Combinatorial Optimization), a CO-specific masked diffusion framework that embeds adaptive expansion as the intrinsic decoding principle. During training, a time-agnostic GNN denoiser learns to reconstruct ground-truth solutions from corrupted partial states, enforcing optimization consistency across different noise levels. At inference, NExCO adopts a Native Adaptive Expansion (NAE) strategy: the denoiser produces confidence scores, candidate sets are formed accordingly, and feasibility projection ensures that selected variables satisfy problem constraints. Through iterative refinement, partial solutions progressively evolve into complete feasible ones. In contrast to COExpander (Ma et al., 2025), which implements AE as an external wrapper around global predictors, NExCO integrates it directly into the diffusion process, achieving the efficiency of GP, the feasibility of LC, and the adaptivity of AE in a unified framework. **The contribution of this paper are:**

**1)** We revisit the Adaptive Expansion (AE) paradigm and point out that existing implementations (e.g., COExpander (Ma et al., 2025)) are merely wrappers around GP predictors, leaving their performance bounded by the backbone and their inference complexity scaling as $O(D_s \cdot C_{GP})$.

**2)** We propose **NExCO**, a masked diffusion framework that natively realizes the AE paradigm by coupling a CO-specific corruption process with a time-agnostic denoiser and introducing the Native Adaptive Expansion (NAE) inference strategy. In doing so, NExCO embeds feasibility, global awareness, and adaptivity directly into the generative process.

**3)** Extensive experiments on representative CO problems (MIS, TSP, CVRP) show that NExCO consistently surpasses prior state-of-the-art solvers in both solution quality and inference efficiency.

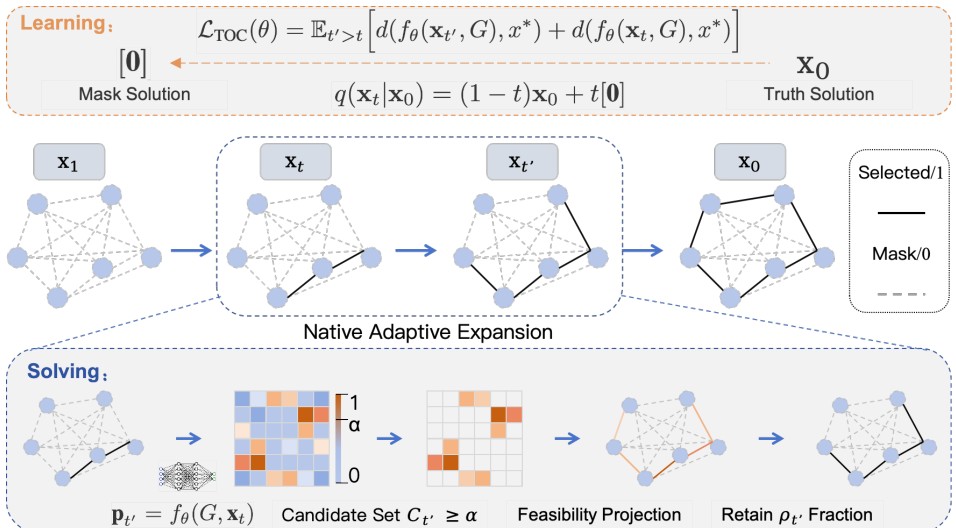

Figure 1: **Overview of NExCO.** Given a graph instance $G$, a ground-truth solution $\mathbf{x}_0 \in \{0,1\}^N$ is corrupted by masking only selected variables (1s) to 0s while keeping unselected ones (0s), producing a partial solution $\mathbf{x}_t$. A time-agnostic GNN denoiser $f_\theta$ predicts confidence scores for all variables without timestep conditioning. During inference, NExCO performs *Native Adaptive Expansion (NAE)*: starting from an fully masked solution $\mathbf{x}_1 = [\mathbf{0}]$, the model progressively unmasks high-confidence variables while a problem-specific projector $\Gamma(\cdot)$ enforces feasibility. This process yields valid intermediate partial solutions and converges to a complete feasible solution.

## 2 PRELIMINARIES AND RELATED WORK

### 2.1 COMBINATORIAL OPTIMIZATION ON GRAPHS

Following standard formulations in neural CO (Sun & Yang, 2023; Li et al., 2023b;a; 2024; Ma et al., 2025; Chen et al., 2024), we represent a problem instance as a graph $G(V, E)$, where $V$ and $E$ denote node and edge sets, respectively, and let $n = |V|$ denote the number of nodes. Decision variables are binary vectors $\mathbf{x} \in \{0,1\}^N$: for *edge-selection* problems, $N = n^2$ and $\mathbf{x}_{i\cdot n+j} = 1$ indicates whether edge $(i, j)$ is selected; for *node-selection* problems, $N = n$ and $\mathbf{x}_i = 1$ indicates whether node $i$ is selected. The feasible region $\Omega$ encodes hard constraints, and the objective is

$$\min_{\mathbf{x} \in \{0,1\}^N} l(\mathbf{x}; G) \quad \text{s.t. } \mathbf{x} \in \Omega. \tag{1}$$

We study three canonical NP-hard tasks: **TSP**: find a minimum-weight Hamiltonian cycle in a complete graph; **MIS**: find a maximum-cardinality independent set; **CVRP**: minimize routing cost subject to degree and vehicle-capacity constraints.

### 2.2 DIFFUSION SOLVERS FOR CO

Diffusion models define a forward corruption $q(\mathbf{x}_t|\mathbf{x}_{t-1})$ and a reverse denoising $\mathbf{p}_\theta(\mathbf{x}_{t-1}|\mathbf{x}_t)$. For binary CO, a natural adaptation is *uniform bit-flip diffusion* (Sun & Yang, 2023; Li et al., 2023b; 2024; 2025; Chen et al., 2026):

$$q(\mathbf{x}_t|\mathbf{x}_{t-1}) = (1 - \beta_t) \mathbf{x}_{t-1} + \beta_t(1 - \mathbf{x}_{t-1}), \tag{2}$$

where $\beta_t \in (0, 1)$ denotes the corruption rate at step $t$. The corresponding $t$-step marginal is

$$q(\mathbf{x}_t|\mathbf{x}_0) = (1 - \bar{\beta}_t) \mathbf{x}_0 + \bar{\beta}_t(1 - \mathbf{x}_0), \tag{3}$$

where $\bar{\beta}_t = 1 - \prod_{s=1}^t (1 - \beta_s)$ is the cumulative corruption rate up to step $t$. Denoisers are then trained to recover the clean signal $\mathbf{x}_0$ or directly predict the optimal solution $x^*$ using variants of likelihood or optimization-consistency losses (Sun & Yang, 2023; Li et al., 2024).

## 2.3 MASKED DIFFUSION

Recent progress has shown that *Masked Diffusion Models (MDMs)* often achieve better performance than uniform bit-flip diffusion in sequence inference tasks (Austin et al., 2021; Ou et al., 2025; Nie et al., 2025b). Instead of symmetrically flipping 0 and 1, MDMs corrupt data by replacing entries with a dedicated [MASK] token, following a continuous trajectory parameterized by $t \in [0, 1]$:

$$q(\mathbf{x}_t|\mathbf{x}_0) = (1 - t) \cdot \mathbf{x}_0 + t \cdot [\text{MASK}]. \tag{4}$$

In the context of combinatorial optimization, solutions are represented as binary vectors $\mathbf{x} \in \{0, 1\}^N$, where each entry (1) corresponds to a selected edge or node. Thus, any state where only a subset of 1s is visible can naturally be interpreted as a *partial solution*. MDMs exploit this perspective: observed entries remain fixed while masked entries are left to be predicted by the denoiser. Moreover, since the corruption level is encoded in the fraction of masked variables rather than in an explicit timestep, training can be made *time-agnostic*. These properties have proven especially useful in large language models, enabling schedule-invariant training and efficient parallel decoding (Zheng et al., 2025; Ou et al., 2025; Nie et al., 2025a; Wu et al., 2025; Sun et al., 2025).

## 3 METHOD

We introduce **NExCO**, a masked diffusion framework tailored for combinatorial optimization. The core idea is to reinterpret the diffusion trajectory in line with the AE paradigm, as a constructive process over *partial solutions* with adaptive expansion, rather than as the generation of probability heatmaps. First, we design a *CO-specific corruption* (§3.1) that masks out selected variables but never introduces false positives, ensuring that intermediate states remain aligned with the feasible manifold. Then, we propose a *time-agnostic graph denoiser* (§3.2) trained with a new *optimization consistency* principle (§3.3), which enforces that all corrupted states of an instance consistently map to the same optimum. Finally, we develop a *native adaptive expansion* decoding strategy (§3.4), which progressively expands partial states into complete solutions under feasibility projection.

### 3.1 FORWARD PROCESS: CO-SPECIFIC CORRUPTION

**Why uniform bit-flip fails to leverage intermediate states.** Prior GP-style diffusion solvers for CO (e.g., DIFUSCO (Sun & Yang, 2023), T2T (Li et al., 2023b), Fast T2T (Li et al., 2024)) adopt *uniform bit-flip* corruption with the marginal formulation in Eq. 3, where each variable is flipped independently with probability $\bar{\beta}_t$. This design causes a fundamental *structural misalignment*: symmetric flipping ignores the combinatorial constraints encoded in the graph. As a result, the corrupted state $\mathbf{x}_t$ quickly drifts away from the feasible manifold. For example, in TSP it contains many edges that violate degree constraints or form subtours (see Fig.1 in (Li et al., 2024)). Such noisy states cannot be interpreted as valid *partial solutions*, but only as dense heatmaps detached from feasibility. Consequently, the denoiser is trained on spurious patterns, and the intermediate trajectory becomes unusable for constructive decoding, forcing prior solvers to discard it and rely solely on heuristic decoding at the final step. This highlights the need for a CO-aware corruption mechanism that *preserves sparsity and respects structural constraints*, so that intermediate states are meaningful partial solutions and can be directly exploited for adaptive expansion.

**Mask diffusion.** A natural alternative to uniform flipping is the *masked diffusion model (MDM)*(Austin et al., 2021; Nie et al., 2025b), which introduces a third state [MASK] and requires the denoiser to reconstruct both 0s and 1s. While effective in language tasks, directly applying MDM to CO is problematic. Take MIS as an example: the validation cost should increase during training, since the goal is to enlarge the independent set. However, as shown in Fig. 2, three-state MDM instead shows a *decreasing* validation cost. This mismatch stems from two factors. First, CO solutions are highly imbalanced: most variables are 0, so the training signal is dominated by negative examples, biasing the denoiser toward pre-

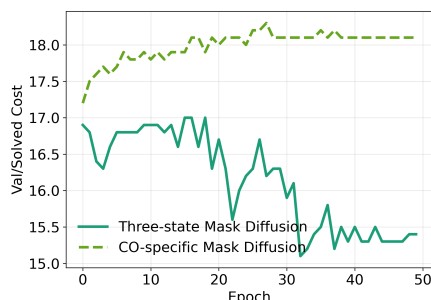

Figure 2: Validation cost curves on MIS.

dicting 0s and shrinking the independent set. Second, unlike TSP where a global Hamiltonian cycle

provides strong priors, MIS depends mainly on local adjacency constraints. Starting from a fully masked state, the model tends to favor "safe" predictions (0s) over "risky" ones (1s that might violate independence), further reinforcing conservative behavior.

**CO-specific corruption.** To address this issue, we unify the `[MASK]` and 0 states into a single *background state* (numerically represented as 0), leaving only `[BACKGROUND]` $\leftrightarrow$ 1 transitions. Intuitively, in CO most variables are fixed at 0 by problem-specific constraints such as degree limits in TSP, capacity restrictions in CVRP, or adjacency rules in MIS. Thus, treating `[MASK]` as distinct provides no additional signal but only exacerbates the imbalance by multiplying negative examples. Under this unification, the forward process becomes a one-way corruption in which only active entries (1s) may be dropped to `[BACKGROUND]`:

$$q(\mathbf{x}_t|\mathbf{x}_0) = (1 - t) \cdot \mathbf{x}_0 + t \cdot \mathbf{0}, \tag{5}$$

where $\mathbf{0}$ denotes the background state. This eliminates the imbalance-driven conservatism of MDM and ensures that intermediate states correspond to valid *partial solutions* aligned with combinatorial feasibility. As shown in Fig. 2, our CO-specific corruption follows the correct trend of increasing validation cost, providing a clean foundation for the time-agnostic denoiser (§3.2) and the Native Adaptive Expansion decoding scheme (§3.4).

## 3.2 TIME-AGNOSTIC DENOISER

**From timestep to mask conditioning.** In prior CO diffusion solvers (Sun & Yang, 2023; Li et al., 2024), the denoiser $f_\theta(\mathbf{x}_t, t, G)$ explicitly conditions on timestep $t$, since $t$ encodes the corruption intensity. This dependence rigidly ties the model to a predefined schedule, limiting generalization across horizons. In our CO-specific corruption process, however, the corruption level is directly *visible in the mask pattern itself*: the fraction of surviving 1s naturally reflects the signal-to-noise ratio. We therefore remove timestep embeddings and design a *time-agnostic denoiser* $f_\theta(\mathbf{x}_t, G)$, which depends only on the corrupted state and the graph instance. This eliminates schedule sensitivity and shifts the focus to structural dependencies and the semantics of partial solutions.

**Model architecture.** We instantiate $f_\theta$ as an anisotropic graph neural network (GNN) (Joshi et al., 2021; Sun & Yang, 2023). Nodes and edges are annotated with task-specific features (e.g., coordinates in TSP, adjacency in MIS, or capacities in CVRP), while $\mathbf{x}_t$ is encoded as binary attributes. Message passing aggregates both structural and partial-solution context, and attention-based pooling captures long-range dependencies. The output is a probability vector $\mathbf{p} \in [0,1]^N$, where $p^{(i)}$ estimates the likelihood of variable $i$ belonging to the optimal solution $x^*$. Compared with conventional denoisers, the only change is the removal of timestep embeddings, highlighting that schedule awareness is unnecessary under mask corruption. Further details are provided in Appendix C.

**Analogy to large language diffusion models.** Our time-agnostic design parallels the success of masked diffusion in large language models (Austin et al., 2021; Ou et al., 2025; Nie et al., 2025b; Wu et al., 2025). In those settings, random masking without explicit timestep conditioning enables scalable pretraining and efficient parallel decoding. By extending this principle from token sequences to graph-structured CO problems, we show that time-agnostic denoising is equally effective when intermediate states correspond to valid partial solutions.

## 3.3 TRAINING OBJECTIVE: TIME-AGNOSTIC OPTIMIZATION CONSISTENCY

**Consistency principle.** Consistency models (Song et al., 2023) learn direct mappings from noisy to clean data, enforcing that predictions across different corruption levels remain stable. Fast T2T (Li et al., 2024) adapted this idea to CO with *optimization consistency*, requiring all corrupted states of an instance to map to its optimal solution $x^*$. This ties denoising directly to the optimization goal, thereby improving one-step prediction quality.

**Time-agnostic optimization consistency.** Our CO-specific corruption is monotone: as $t$ increases, supports shrink, so for any $0 < t < t' < 1$ we have $\text{supp}(\mathbf{x}_{t'}) \subseteq \text{supp}(\mathbf{x}_t)$. Each corrupted state $\mathbf{x}_t$ is thus a valid subset of $x^*$. This property allows us to drop timestep embeddings and directly enforce consistency across corruption levels. Formally, the *time-agnostic optimization consistency* loss is

$$\mathcal{L}_{\text{TOC}}(\theta) = \mathbb{E}_{t'>t}\Big[d(f_\theta(\mathbf{x}_{t'}, G), x^*) + d(f_\theta(\mathbf{x}_t, G), x^*)\Big], \tag{6}$$

where $d(\cdot, \cdot)$ is binary cross-entropy or a task-specific discrepancy. Training under $\mathcal{L}_{\text{TOC}}$ amounts to reconstructing $x^*$ from multiple *partial solutions*. Because the corruption never introduces false positives, every $\mathbf{x}_t$ stays close to the feasible manifold, providing supervision that is inherently aligned with CO constraints. This contrasts with uniform diffusion, where intermediate states are often unrealistic and force the model to correct artifacts.

**Practical note.** The reference solution $x^*$ can come from exact solvers on small/medium instances or high-quality heuristics on larger ones. The TOC loss remains valid in both cases, as it only requires a consistent reference per instance. We empirically confirm this in Table 3, where models trained on suboptimal labels still deliver competitive solutions.

## 3.4 INFERENCE PROCESS: NATIVE ADAPTIVE EXPANSION (NAE)

**Motivation.** Existing diffusion-based CO solvers generate a sequence of $T_s$ global probability heatmaps $\mathbf{p}_1, \ldots, \mathbf{p}_{T_s} \in [0, 1]^N$, but these intermediate states are not semantically valid partial solutions. As a result, prior methods typically exploit only the final step via heuristic decoding:

$$\hat{x} = \Gamma\big(\text{Decode}(\mathbf{p}_{T_s})\big), \tag{7}$$

where Decode is a heuristic (e.g., greedy search) and $\Gamma$ enforces feasibility (Sun & Yang, 2023). This *under-utilization* of the trajectory is a key limitation: despite producing many intermediate states, only a final one-shot prediction is retained. COExpander (Ma et al., 2025) alleviates this via *adaptive expansion* (AE), but AE is realized as an external wrapper around GP predictors. Its complexity depends on both the number of expansion rounds $D_s$ and the per-call cost $C_{\text{GP}}$ of the backbone: $\text{Cost}_{\text{AE}} = \mathcal{O}(D_s \cdot C_{\text{GP}})$. Thus, wrapping diffusion solvers (e.g., Fast T2T (Li et al., 2024)) yields $\mathcal{O}(D_s \cdot T_s)$ complexity, while wrapping GCN (Joshi et al., 2019) reduces it to $\mathcal{O}(D_s)$ but still ties performance to external heatmap quality (Xia et al., 2024).

By contrast, our framework produces semantic partial solutions along the diffusion trajectory. This enables *Native Adaptive Expansion (NAE)*: a deterministic expansion procedure that reuses the denoiser once per stage and enforces feasibility at each step, achieving $\mathcal{O}(T_s)$ complexity while making full constructive use of the entire trajectory.

**Procedure.** NAE begins from $\mathbf{x}_1$ as fully masked solution $\mathbf{0}$ and expands iteratively. At step $t$, the denoiser outputs a confidence vector $\mathbf{p}_t = f_\theta(G, \mathbf{x}_{t-1})$. Variables above threshold $\alpha$ form a candidate set $\mathcal{C}_t$, which is projected onto a feasible subset $\mathcal{S}_t$ by $\Gamma(\cdot)$. From this subset, a fraction $\rho_t$ of entries is selected, where $\rho_t$ may be set as a tunable hyperparameter or determined by an evenly spaced schedule over steps. Repeating this process yields a trajectory of feasible partial solutions until completion, as summarized in Algorithm 1.

---

**Algorithm 1:** Native Adaptive Expansion (NAE)

**Input:** Graph $G$, denoiser $f_\theta$, iterations $T_s$, threshold $\alpha$, expansion schedule $\{\rho_t\}$.

Initialize $\mathbf{x}_1 \leftarrow \mathbf{0}$;

**for** $t = 1, \ldots, T_s$ **do**
  $\mathbf{p}_t \leftarrow f_\theta(G, \mathbf{x}_{t-1})$;
  $\mathcal{C}_t \leftarrow \{i \mid \mathbf{x}_{t-1}^{(i)} = 0, \mathbf{p}_t^{(i)} \geq \alpha\}$;
  Project candidates: $\mathcal{S}_t \leftarrow \Gamma(\mathcal{C}_t, \mathbf{p}_t, \mathbf{x}_{t-1})$;
  Retain $\rho_k$ fraction of $\mathcal{S}_t$ ;
  Update $\mathbf{x}_t$ by activating retained entries;

**return** $\mathbf{x}_{T_s}$

---

**Feasibility projection.** Although different CO tasks impose different feasibility rules, the projection operator $\Gamma(\cdot)$ follows a single task-agnostic template across all problems we study. At each expansion step, the model produces a confidence vector $\mathbf{p}$, and $\Gamma(\cdot)$ constructs the next partial solution using the same three-stage procedure. First, candidate variables are sorted in descending confidence. Second, candidates are examined sequentially. Third, a candidate is activated only when doing so satisfies a simple boolean feasibility predicate. This local predicate is lightweight to compute and varies only in its constraint definition, not in the mechanism of feasibility projection. Consequently, extending NAE to a new CO task requires only defining this predicate, while the entire three-stage projection pipeline remains intact. Formally, the projection step solves:

$$\mathcal{S}_t = \arg\max_{x^{(i)} \subseteq \mathcal{C}_t, \, x \in \Omega} \sum_i p_t^{(i)} x^{(i)}, \tag{8}$$

which is implemented by inserting candidates in descending confidence order and accepting them only when the feasibility predicate holds. In practice, this yields a simple and uniform instantiation

Table 2: Results on synthetic TSP problem instances. BS: Beam Search.

| Algorithm | Type | TSP-100 | | | TSP-500 | | | TSP-1000 | | |
|---|---|---|---|---|---|---|---|---|---|---|
| | | Length↓ | Drop↓ | Time | Length↓ | Drop↓ | Time | Length↓ | Drop↓ | Time |
| *Mathematical Solvers or Heuristics* | | | | | | | | | | |
| Concorde (Applegate et al., 2006) | Exact | 7.76* | – | 0.23s | 16.55* | – | 18.65s | 23.12* | – | 84.38s |
| LKH3 (512) (Helsgaun, 2017) | Heuristics | 7.76 | 0.00% | 0.12s | 16.55 | 0.00% | 1.17s | 23.12 | 0.01% | 2.91s |
| *Learning-based Solvers with Greedy Decoding* | | | | | | | | | | |
| AM +BS (Kool et al., 2019) | LC | 7.95 | 2.48% | 0.61s | 19.53 | 18.03% | 1.31s | 29.90 | 29.24% | 5.91s |
| BQ-NCO +BS (Drakulic et al., 2023) | LC | 7.76 | 0.01% | 0.19s | 16.64 | 0.55% | 7.03s | 23.47 | 1.38% | 17.81s |
| GCN +BS (Joshi et al., 2019) | GP | 8.41 | 8.38% | 0.28s | 30.37 | 83.55% | 17.81s | 51.26 | 121.73% | 24.23s |
| DIMES (Qiu et al., 2022) | GP | 8.01 | 3.23% | 0.06s | 17.17 | 3.74% | 0.45s | 24.79 | 7.22% | 1.12s |
| DIFUSCO ($T_S = 50$) (Sun & Yang, 2023) | GP | 7.78 | 0.26% | 0.59s | 16.82 | 1.64% | 1.43s | 23.57 | 1.94% | 5.04s |
| T2T ($T_s = 50, T_g = 30$) (Li et al., 2023b) | GP | 7.76 | 0.07% | 1.34s | 16.68 | 0.82% | 3.05s | 23.44 | 1.40% | 9.23s |
| Fast T2T ($T_s = 5$) (Li et al., 2024) | GP | 7.76 | 0.08% | 0.06s | 16.72 | 1.02% | 0.27s | 23.38 | 1.12% | 0.99s |
| Fast T2T ($T_s = 5, T_g = 25$) (Li et al., 2024) | GP | 7.76 | 0.03% | 0.31s | 16.61 | 0.39% | 1.41s | 23.25 | 0.58% | 5.81s |
| COExpander ($D_s = 3, T_s = 5$) (Ma et al., 2025) | AE | 7.76 | 0.04% | 0.18s | 16.63 | 0.52% | 0.61s | 23.34 | 0.95% | 2.26s |
| NExCO ($D_s = 3$) | NAE | 7.76 | 0.04% | 0.05s | 16.61 | 0.39% | 0.23s | 23.31 | 0.85% | 0.91s |
| NExCO ($D_s = 5$) | NAE | 7.76 | 0.03% | 0.08s | 16.59 | 0.28% | 0.33s | 23.26 | 0.63% | 1.31s |
| NExCO ($D_s = 7$) | NAE | 7.76 | 0.02% | 0.11s | 16.59 | 0.25% | 0.43s | 23.24 | 0.52% | 1.68s |
| *Learning-based Solvers with $4\times$ Sampling Decoding* | | | | | | | | | | |
| LEHD PRC 100 (Luo et al., 2023) | LC | 7.76 | 0.01% | 0.64s | 16.61 | 0.34% | 3.75s | 23.44 | 1.22% | 20.16s |
| Fast T2T ($T_s = 5, T_g = 5$) (Li et al., 2024) | GP | 7.76 | 0.01% | 0.99s | 16.58 | 0.21% | 5.16s | 23.22 | 0.42% | 17.42s |
| COExpander ($D_s = 3, T_s = 5$) (Ma et al., 2025) | AE | 7.76 | 0.01% | 0.61s | 16.59 | 0.24% | 2.21s | 23.27 | 0.64% | 8.43s |
| NExCO ($D_s = 5$) | NAE | 7.76 | 0.01% | 0.25s | 16.57 | 0.14% | 1.16s | 23.20 | 0.35% | 4.85s |

across tasks: TSP: edges are added while maintaining degree-2 and subtour-free constraints; MIS: a vertex is selected only if all neighbors remain inactive; CVRP: routing edges are inserted provided vehicle-capacity constraints are not violated. Candidates that fail the predicate are skipped, preventing conflicts and ensuring that $\mathcal{S}_t \subseteq \mathcal{C}_t$ is always a feasible and high-confidence partial solution.

**Convergence and complexity.** We provide a formal convergence analysis of the NAE procedure. Because the forward corruption in our CO-specific diffusion process is one-way absorbing $(1 \rightarrow 0)$, every diffusion state remains a feasible partial solution. The reverse step expands this partial solution via $\mathbf{x}_{t+1} = \Gamma(\mathbf{x}_t \vee \mathbf{z}_t)$, where $\mathbf{z}_t$ is the candidate activation mask predicted by the denoiser, and $\Gamma$ is a feasibility projector. We assume the following mild and standard conditions, satisfied by the projectors $\Gamma(\cdot)$ used for TSP, MIS, and CVRP:

**(A1) Monotone projection:** $\Gamma(\mathbf{x}) \succeq \mathbf{x}$ for all feasible $\mathbf{x}$.

**(A2) Strict expandability:** $\exists \mathbf{z}_t$ such that $\Gamma(\mathbf{x}_t \vee \mathbf{z}_t) \succ \mathbf{x}_t$ whenever $\mathbf{x}_t$ is incomplete.

**(A3) Bounded solution size:** Any complete feasible solution contains at most $N_{\max}$ active variables.

**Proposition 1 (Finite-time convergence of NAE)** *Under assumptions (A1)-(A3), NAE generates a monotone sequence $\mathbf{x}_0 \preceq \mathbf{x}_1 \preceq \cdots$ and converges to a complete feasible solution in at most $N_{\max}$ iterations.*

**Remark.** The upper bound $N_{\max}$ is fully consistent with typical CO structures:

- TSP: $N_{\max} = N$ edges in a Hamiltonian tour.
- MIS: $N_{\max} \leq n$ selected nodes.
- CVRP: $N_{\max}$ equals the total number of edges across all valid routes.

This analysis formalizes the intuition that NAE is a monotone constructive decoder that reaches a complete feasible solution in finite time. In terms of efficiency, NAE requires $\mathcal{O}(T_s)$ denoiser calls, matching the order of diffusion while being asymptotically more efficient than COExpander, whose wrapper design incurs $\mathcal{O}(D_s \cdot T_s)$ complexity.

## 4 EXPERIMENTS

### 4.1 EXPERIMENTS ON TSP

**Datasets.** Each TSP instance consists of $N$ two-dimensional coordinates and a reference optimal solution. Following standard practice (Sun & Yang, 2023), we generate instances by uniformly

sampling $N$ nodes from the unit square $[0,1]^2$. The training sets contain 1,280K, 128K, and 64K instances for TSP-100, TSP-500, and TSP-1000, respectively. The corresponding test sets consist of 1,280 instances for TSP-100 and 128 instances each for TSP-500 and TSP-1000. Reference solutions are obtained using Concorde (Applegate et al., 2006). We further include large-scale TSP-10K instances and real-world TSPLIB benchmarks, with results reported in Appendix B.2 and B.3.

**Metrics.** We evaluate solvers on three metrics: 1) *Length*: average tour length of the produced solutions; 2) *Drop*: relative deviation from the reference solution; 3) *Time*: average time per instance.

**Setting.** For NExCO, we vary the number of expansion rounds $D_s \in \{3, 5, 7\}$ in the NAE procedure to balance solution quality and runtime. The confidence threshold $\alpha$ is tuned as a hyperparameter. For diffusion-based baselines, we adopt standard configurations (Li et al., 2023b): $T_s$ denotes the number of inference steps, and $T_g$ denotes the number of gradient refinement steps. Unless otherwise stated, all methods employ greedy decoding with an optional 2-Opt heuristic in post-processing.

**Main results.** Table 2 reports the comparison across different scales. NExCO consistently surpasses state-of-the-art learning-based solvers in both solution quality and runtime. On TSP-100, Fast T2T achieves a $0.03\%$ gap in 0.31s, whereas NExCO attains the same gap in only 0.08s, yielding a $3.9\times$ speedup. Similarly, on TSP-500, NExCO reduces the gap to $0.25\%$ within 0.43s, while Fast T2T requires 1.41s to reach $0.39\%$, corresponding to both a $1.5\times$ improvement in gap and a $3\times$ speedup. Taken together, these results demonstrate that embedding adaptive expansion natively into diffusion not only accelerates inference by 2–4×, but also consistently reduces optimality gaps, highlighting the effectiveness of NExCO as a next-generation neural TSP solver.

**Ablation on reference quality.** We further examine the robustness of NExCO when trained with suboptimal supervision. To this end, we construct perturbed labels by applying 2-Opt local search to the ground-truth solutions. As shown in Table 3, these perturbed references are

Table 3: Ablation on reference quality.

| Label / Model | Length ↓ | Drop ↓ | Length ↓ | Drop ↓ |
|---|---|---|---|---|
| 2-Opt Perturbation | 16.82 | 1.65% | 17.11 | 3.35% |
| NExCO ($D_s = 5$) | 16.60 | 0.31% | 16.60 | 0.31% |
| NExCO ($D_s = 7$) | 16.59 | 0.25% | 16.59 | 0.26% |

significantly worse than the exact optima, with gaps of $1.65\%$ and $3.35\%$ on TSP-500. Nevertheless, models trained on such labels still deliver highly competitive results, achieving final gaps of only $0.25\% - 0.31\%$. This demonstrates that NExCO is not tied to exact optimal labels but can effectively leverage high-quality heuristic solutions as consistent training signals, making it broadly applicable in practical CO scenarios where exact optima are often unavailable.

**Generalization study.** Table 4 reports cross-scale transfer results. Models trained on small instances generalize poorly, with DIFUSCO, T2T, and Fast T2T all incurring $\sim 3\%$ gaps on TSP-1000. Training on medium or large instances improves robustness, and NExCO shows the strongest transferability. In particular, when trained on TSP-500, it generalizes to TSP-1000 with only $0.58\%$ gaps. Similarly, the model trained on TSP-1000 achieves a $0.37\%$ gap on TSP-500, surpassing baselines that are directly trained on TSP-500. These results highlight that native adaptive expansion yields consistently superior cross-scale generalization compared with existing diffusion-based solvers.

Table 4: Cross-scale generalization results on TSP. Each entry reports Length, Gap (%).

| Training / Testing | TSP-100 | TSP-500 | TSP-1000 |
|---|---|---|---|
| DIFUSCO ($T_s$=50) | 7.78, 0.23% | 7.85, 1.16% | 7.87, 1.42% |
| T2T ($T_s$=50, $T_g$=30) | 7.77, 0.08% | 7.95, 2.47% | 7.91, 1.96% |
| Fast T2T ($T_s$=5, $T_g$=5) | 7.77, 0.02% | 7.79, 0.40% | 7.80, 0.55% |
| NExCO ($D_s = 5$) | 7.76, 0.02% | 7.77, 0.18% | 7.81, 0.72% |
| DIFUSCO ($T_s$=50) | 17.05, 3.04% | 16.78, 1.40% | 16.86, 1.85% |
| T2T ($T_s$=50, $T_g$=30) | 16.92, 2.25% | 16.68, 0.81% | 16.72, 1.00% |
| Fast T2T ($T_s$=5, $T_g$=5) | 16.92, 1.77% | 16.61, 0.38% | 16.63, 0.51% |
| NExCO ($D_s = 5$) | 17.04, 2.96% | 16.59, 0.27% | 16.60, **0.37%** |
| DIFUSCO ($T_s$=50) | 24.04, 3.98% | 23.65, 2.30% | 23.63, 2.21% |
| T2T ($T_s$=50, $T_g$=30) | 23.85, 3.16% | 23.47, 1.51% | 23.41, 1.23% |
| Fast T2T ($T_s$=5, $T_g$=5) | 23.77, 3.08% | 23.31, 0.81% | 23.25, 0.58% |
| NExCO ($D_s = 5$) | 24.01, 3.86% | 23.25, **0.58%** | 23.25, 0.57% |

## 4.2 EXPERIMENTS ON MIS.

**Datasets.** Following (Sun & Yang, 2023; Qiu et al., 2022; Li et al., 2018; Ahn et al., 2020), we consider two benchmark datasets: RB graphs and Erdős–Rényi (ER) graphs. For RB graphs, we generate small- and large-scale instances by sampling the number of vertices from $[200, 300]$ and $[800, 1200]$, respectively. For ER graphs, we construct random graphs with 700–800 nodes, where each edge is added independently with probability $0.15$. In total, we generate 90,000 small and 6,400 large RB instances for training, with 500 test instances. For ER graphs, 163,840 instances are used for training and 500 test instances are adopted from (Qiu et al., 2022).

Table 5: Results on synthetic MIS problem instances.

| Algorithm | Type | RB-[200-300] | | | RB-[800-1200] | | | ER-[700-800] | | |
|---|---|---|---|---|---|---|---|---|---|---|
| | | Size↑ | Drop↓ | Time | Size↑ | Drop↓ | Time | Size↑ | Drop↓ | Time |
| KaMIS (Lamm et al., 2016) | Heuristics | 20.10 | – | 45.81s | 43.00 | – | 56.97s | 44.87 | – | 48.49s |
| DIMES (Qiu et al., 2022) | GP | – | – | – | – | – | – | 38.24 | 14.78% | 2.8s |
| DIFUSCO ($T_s = 100$) (Sun & Yang, 2023) | GP | 18.52 | 7.81% | 1.9s | – | – | – | 37.03 | 18.53 | 2.58s |
| T2T ($T_s = 50, T_g = 30$) (Li et al., 2023b) | GP | 18.98 | 5.49% | 2.52s | – | – | – | 39.81 | 11.28 | 3.33s |
| Fast T2T ($T_s = 5, T_g = 5$) (Li et al., 2024) | GP | 19.58 | 2.54% | 0.39s | 39.34 | 8.51% | 2.76s | 40.78 | 9.31% | 1.22s |
| COExpander ($D_s = 20, T_s = 1$) (Ma et al., 2025) | AE | 19.60 | 2.39% | 0.20s | 41.09 | 4.36% | 2.05s | 42.44 | 5.62% | 1.53s |
| NExCO ($D_s = 5$) | NAE | 19.70 | 1.97% | 0.11s | 40.94 | 4.79% | 0.75s | 42.64 | 4.96% | 0.44s |
| NExCO ($D_s = 7$) | NAE | 19.76 | 1.66% | 0.14s | 41.25 | 4.07% | 1.00s | 42.98 | 4.20% | 0.56s |

Table 6: Results on synthetic CVRP problem instances.

| Algorithm | Type | CVRP 50 | | | CVRP 100 | | | CVRP 200 | | |
|---|---|---|---|---|---|---|---|---|---|---|
| | | Length↓ | Drop↓ | Time | Length↓ | Drop↓ | Time | Length↓ | Drop↓ | Time |
| HGS (Helsgaun, 2017) | Heuristics | 10.37 | – | 1.01s | 15.56 | – | 20.03s | 19.63 | – | 60.02s |
| Sym-NCO (Kim et al., 2022) | LC | 10.57 | 1.95% | 0.09s | 15.93 | 2.37% | 0.19s | 20.19 | 2.86% | 0.36s |
| COExpander ($D_s = 3, T_s = 1$) (Ma et al., 2025) | AE | 10.77 | 3.85% | 0.03s | 16.22 | 4.19% | 0.06s | 20.52 | 4.58% | 0.20s |
| NExCO ($D_s$=3) | NAE | 10.48 | 1.12% | 0.04s | 15.83 | 1.73% | 0.08s | 20.18 | 2.76% | 0.27s |
| NExCO ($D_s$=5) | NAE | 10.46 | 0.85% | 0.06s | 15.78 | 1.40% | 0.12s | 20.11 | 2.45% | 0.39s |

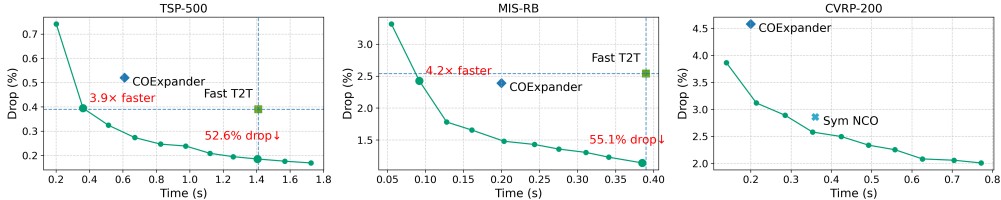

Figure 3: **Effect of expansion steps on performance.** Green curves show our method with different numbers of expansion steps. Increasing steps improves solution quality (smaller drop) while linearly increasing inference time. Reference markers denote SOTA baselines (Fast T2T, COExpander, Sym-NCO), against which our method achieves consistently better efficiency–quality trade-offs.

**Metrics.** 1) *Size*: the average solution size; 2) *Drop*: the relative deviation from the reference solution obtained by KaMIS (Lamm et al., 2016); 3) *Time*: the average runtime per instance.

**Main results.** Table 5 shows that NExCO consistently improves both efficiency and solution quality over diffusion- and expansion-based baselines. On RB-[200–300], NExCO lowers the drop from 2.54% (Fast T2T) to 1.66% while being 2.8× faster. On ER, it achieves a 4.20% drop compared with 9.31% for Fast T2T, with more than 2× speedup. Compared with COExpander, NExCO also attains smaller gaps in much less time: on RB-[800–1200], 4.07% in 1.00s vs. 4.36% in 2.05s. These results demonstrate that NAE not only improves scalability but also yields consistently higher-quality solutions in MIS. Additional generalization results are provided in Appendix B.1.

## 4.3 EXPERIMENTS ON CVRP

**Datasets.** Each CVRP instance consists of a depot, $N$ customer coordinates, and corresponding demands. Coordinates are uniformly sampled from the unit square $[0, 1]^2$, demands are drawn as integers from $[1, 10]$, and vehicle capacities are fixed to 40, 50, and 80 for CVRP-50, CVRP-100, and CVRP-200, respectively. The training sets contain 1,280K, 640K, and 32K instances for CVRP-50/100/200, while the test sets include 10K instances for CVRP-50/100 and 100 instances for CVRP-200. Reference solutions are obtained using the HGS solver (Helsgaun, 2017).

**Main results.** As summarized in Table 6, NExCO achieves consistent improvements across all scales. Compared with LC methods such as Sym-NCO (Kim et al., 2022), which yield moderate drops, and AE-based COExpander, which suffers from even larger deviations, NExCO substantially reduces the gap while also cutting runtime. For example, it improves from 3.86% → 0.85% on CVRP-50, 4.16% → 1.40% on CVRP-100, and 4.84% → 2.45% on CVRP-200, corresponding to 1.6 − 2× speedup. Notably, GP-based diffusion solvers have not been extended to CVRP due to its intricate capacity constraints, further highlighting the strength of native adaptive expansion.

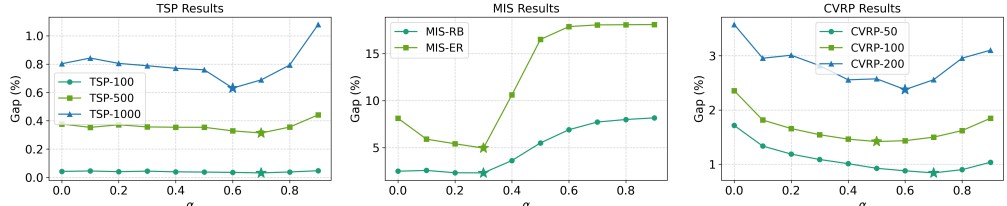

Figure 4: **Effect of the candidate threshold** $\alpha$**.** Performance across TSP, MIS, and CVRP improves at moderate thresholds, but deteriorates when $\alpha$ is set too high (restrictive) or too low (noisy).

## 4.4 HYPERPARAMETER STUDY

We analyze two key hyperparameters in NExCO: the number of expansion steps and the candidate threshold $\alpha$. As shown in Fig. 3, increasing the number of expansion steps consistently improves solution quality while runtime grows nearly linearly, revealing a clear trade-off between efficiency and performance. This allows practitioners to flexibly adjust inference cost, and NExCO remains superior to strong baselines (Fast T2T, COExpander, SymNCO) even under reduced step budgets. Similarly, Fig. 4 shows that moderate values of $\alpha$ yield the best results: overly high thresholds become too restrictive, while overly low thresholds admit noisy candidates. Importantly, the optimal range of $\alpha$ is largely shared across all sizes and distributions within each task (e.g., TSP-100/500/1000; MIS-RB/ER; CVRP-50/100/200), demonstrating strong task-level robustness. This cross-setting consistency means that NExCO requires minimal per-instance tuning, as a single hyperparameter choice generalizes reliably across scales within the same CO task.

## 5 CONCLUSION

In this work, we proposed **NExCO**, a masked diffusion framework that realizes adaptive solution expansion as a native generative principle for neural combinatorial optimization. Our framework is built on three key components: a CO-specific forward corruption that preserves sparsity and yields semantic partial solutions, a time-agnostic GNN denoiser trained under optimization consistency, and a Native Adaptive Expansion (NAE) inference strategy that progressively selects confident variables under feasibility constraints. This framework has demonstrated the effectiveness across three representative CO problems. We believe this work opens up new opportunities for integrating constructive expansion mechanisms into diffusion-based generative modeling, and provides a step forward toward scalable and general-purpose neural solvers for combinatorial optimization.

## ETHICS STATEMENT

This paper presents a new masked diffusion framework for neural combinatorial optimization. The proposed method addresses fundamental challenges in existing neural solvers, including the inefficiency of local construction, the constraint conflicts of global prediction, and the reliance on external predictors in adaptive expansion. Our contribution is methodological in nature, aiming to improve both solution quality and inference efficiency across benchmark CO problems such as MIS, TSP, and CVRP. We expect this work to benefit the broader research community by providing a more principled foundation for scalable and effective neural solvers in discrete optimization.

We do not anticipate any negative societal impacts arising from this research. Our work does not involve sensitive personal data, human subjects, or applications with immediate ethical risks. Furthermore, we do not identify issues related to conflicts of interest, discrimination or fairness, privacy or security, legal compliance, or research integrity. As a methodological contribution validated on public benchmarks, this work is aligned with the ethical standards of the ML community.

## REPRODUCIBILITY STATEMENT

We provide detailed descriptions of the datasets, evaluation metrics, and hyperparameter choices in Sec. 4. The model architecture, hardware configuration, and complete training details are provided in Appendix C and Appendix D. All source code, preprocessed datasets, and pretrained checkpoints will be publicly released upon publication to ensure full reproducibility.

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

# APPENDIX

# A ADDITIONAL RELATED WORKS

## A.1 LEARNING-BASED COMBINATORIAL OPTIMIZATION

Learning-based combinatorial optimization approaches can be broadly grouped into three categories: constructive methods, improvement-based methods, and divide-and-conquer frameworks.

**Constructive methods** can be further divided into Local Construction (LC) and Global Prediction (GP) paradigms. LC approaches (Kool et al., 2019; Kwon et al., 2020; Kim et al., 2022; Berto et al., 2023; Drakulic et al., 2023; Pan et al., 2025) build solutions sequentially, selecting one variable at a time and ensuring feasibility at each step, but often suffer from myopic decisions and slow decoding. RL4CO community develops a comprehensive repository for this category of methods (Berto et al., 2023). GP approaches (Joshi et al., 2019; Fu et al., 2021; Qiu et al., 2022; Schuetz et al., 2022; Min et al., 2023; Sun & Yang, 2023; Xia et al., 2024; Li et al., 2024; Xiao et al., 2024), by contrast, predict global probability heatmaps in a single forward pass. This makes them efficient and globally aware, but the raw predictions often violate constraints and thus require post-processing to enforce feasibility. Within this paradigm, generative modeling methods (Hottung et al., 2021; Sun & Yang, 2023; Li et al., 2023b; 2024; Zhao et al., 2024; Sanokowski et al., 2024) aim to learn a distribution over high-quality solutions for each instance. Feasible solutions can then be obtained by sampling from this distribution, which has been shown to yield competitive or even superior solver performance. To bridge these trade-offs, COExpander (Ma et al., 2025) recently introduced the *Adaptive Expansion (AE)* paradigm, which interpolates between LC's fine-grained feasibility and GP's global awareness. However, AE in COExpander is implemented only as a wrapper around GP predictors, leaving its effectiveness bounded by the quality and efficiency of the underlying backbone. This limitation motivates our work, which aims to instantiate AE as a native generative principle within a diffusion framework.

**Improvement-based solvers** (Chen & Tian, 2019; Oliveira da Costa et al., 2020; Wu et al., 2021; Sui et al., 2021; Li et al., 2021; Hou et al., 2023; Ma et al., 2023; Li et al., 2025) focus on refining an initial solution, typically generated by a simple heuristic (e.g., greedy construction), through iterative optimization guided by neural networks or local search operators. While these methods can improve solution quality and naturally enforce feasibility at each step, they suffer from two key drawbacks: their dependence on heuristic initializers limits generality across problem settings, and the need for multiple refinement rounds incurs substantial computational overhead.

**Divide-and-conquer (D&C) framerworks** (Fu et al., 2021; Kim et al., 2021; Luo et al., 2023; 2025; Zheng et al., 2024; Ye et al., 2024) address scalability by decomposing large CO instances into smaller subproblems, solving them with either classical heuristics or neural solvers, and then aggregating the partial solutions into a global one. Scalability has long been a central challenge in neural CO: for GP-based methods, obtaining supervision signals is prohibitively expensive; for RL-based sequential models, issues like sparse rewards and unstable training further hinder their applicability to large-scale settings. DC provides a practical alternative to mitigate these limitations, and is largely orthogonal to constructive methods: it can be applied on top of LC, GP, or AE models, enabling them to operate effectively on larger-scale instances through problem decomposition.

## A.2 MASK DIFFUSION MODEL

Diffusion models were first developed for continuous domains with Gaussian transitions (Sohl-Dickstein et al., 2015; Ho et al., 2020; Song et al., 2021), and later extended to discrete spaces by reformulating the forward process as a discrete-state Markov chain (Hoogeboom et al., 2021; Austin et al., 2021; Lou et al., 2024). Among these extensions, the *masked diffusion model (MDM)* has proven particularly effective: instead of arbitrary bit flips, variables are corrupted into an absorbing mask state, yielding semantically meaningful intermediate states. This simple yet powerful design enables principled training (Sahoo et al., 2024; Shi et al., 2024), scales efficiently to large models (Gong et al., 2025; Nie et al., 2025b; Wu et al., 2025; Sun et al., 2025; Ou et al., 2025), and consistently outperforms autoregressive models in reasoning and planning tasks (Ye et al., 2025; Nie et al., 2025a; Zheng et al., 2025). With these advantages, masked diffusion has become the de facto framework for discrete generative modeling, offering interpretable partial states, schedule-invariant

training, and efficient decoding that resonate strongly with the requirements of neural combinatorial optimization.

## B  Additional Experiments

### B.1  Cross-distribution and cross-scale generalization on MIS

Table 7: Cross-distribution and cross-scale generalization results on MIS. Each entry reports the average solution size.

| Algorithm | Cross-Distribution | | | Cross-Scale | |
|---|---|---|---|---|---|
| | $p = 0.2$ | $p = 0.3$ | $p = 0.4$ | [350–400] | [1400–1600] |
| KaMIS | 35.30 | 24.37 | 18.18 | 37.96 | 50.95 |
| DIFUSCO | 26.25 | 15.84 | 11.75 | 27.31 | 34.39 |
| Fast T2T | 29.52 | 17.77 | 13.27 | 32.56 | 36.95 |
| NEXCO | **31.24** | **18.14** | **14.17** | **34.65** | **38.39** |

Table 7 compares the performance of different solvers under distribution shifts and varying graph sizes. On random ER graphs with increasing edge probabilities ($p = 0.2, 0.3, 0.4$), learning-based solvers exhibit a clear gap compared to the heuristic KaMIS, but NExCO consistently achieves larger solution sizes than both DIFUSCO and Fast T2T, e.g., 31.24 vs. 29.52 and 26.25 at $p = 0.2$. A similar trend holds in cross-scale settings: when transferring from RB-[350–400] and RB-[1400–1600], NExCO again outperforms other learning-based solvers, achieving 34.65 and 38.39 respectively, compared to 32.56 and 36.95 from Fast T2T. These results highlight that native adaptive expansion maintains superior generalization under both distributional and scale shifts.

### B.2  Scalability on TSP-10K

To further evaluate scalability, we conducted experiments on the large-scale TSP-10K dataset. We compare NExCO with strong baselines, including the classical heuristic LKH, the divide-and-conquer approach GLOP (Ye et al., 2024), and diffusion-based solvers DIFUSCO (Sun & Yang, 2023), DISCO (Zhao et al., 2024), and Fast T2T (Li et al., 2024). For diffusion-based methods, we follow standard practice (Sun & Yang, 2023; Li et al., 2024) and apply K-Nearest Neighbor (KNN) sparsification with a sparse factor of 100 to restrict the search space by sampling 100 neighbors for each node. As shown in Table 8, NExCO achieves the best balance between quality and efficiency: it reduces the optimality gap to 1.53% in 52s, improving over diffusion baselines (e.g., 1.63% in 70s by Fast T2T) while being substantially faster than LKH. These results highlight the scalability advantage of native adaptive expansion in large-scale settings.

To extend to larger regimes, NEXCO is naturally compatible with standard scaling strategies used in large-graph CO systems, such as divide-and-conquer frameworks (e.g., region partitioning similar to GLOP (Ye et al., 2024)), or replacing the backbone GNN with lightweight sparse-attention Transformers (Luo et al., 2025). These directions do not alter the proposed generative principle and can be incorporated in future work to achieve industrial-scale deployments. In addition, the partial-solution semantics of NEXCO provide a natural mechanism for overcoming the scarcity and high cost of supervision in large-scale CO tasks. Because the model operates on feasible partial states rather than requiring complete high-quality solutions at every step, it can be trained using incomplete, heuristic, or low-cost labels and gradually refine its own predictions through iterative self-training. This significantly reduces reliance on exact solvers, whose computational cost grows prohibitively with instance size. The ability to learn from weak or partial supervision aligns with recent scalable training pipelines (Luo et al., 2025; Li et al., 2025) that couple lightweight backbones with progressive bootstrapping. Incorporating such strategies offers a promising path for applying NEXCO in settings where high-quality ground-truth labels are expensive or unavailable, thereby further extending its applicability to truly large and industrial-scale CO environments.

Table 8: **Results on TSP-10K.** NExCO achieves the best trade-off between solution quality and runtime.

| Method | Type | Length ↓ | Drop ↓ | Time |
|---|---|---|---|---|
| LKH | Heuristics | 71.77 | – | 332s |
| GLOP (more revisions) | DC | 75.29 | 4.90% | 15s |
| DIFUSCO ($T_s = 100$) | GP | 73.91 | 2.98% | 124s |
| DISCO | GP | 73.84 | 2.88% | 92s |
| Fast T2T ($T_s = 5$) | GP | 72.94 | 1.63% | 70s |
| NExCO ($D_s = 5$) | NAE | 72.87 | 1.53% | 52s |

Table 9: Solution quality (%) for methods trained on TSP-100 problems and evaluated on **TSPLIB** instances with 50–200 nodes. * denotes results quoted from previous works (Li et al., 2024).

| Instances | AM* | GCN* | Learn2OPT* | GNNGLS* | DIFUSCO* | T2T* | Fast T2T* | NExCO |
|---|---|---|---|---|---|---|---|---|
| eil51 | 16.767 | 40.025 | 1.725 | 1.529 | 2.82 | 0.14 | 0.00 | 0.00 |
| berlin52 | 4.169 | 33.225 | 0.449 | 0.142 | 0.00 | 0.00 | 0.00 | 0.00 |
| st70 | 1.737 | 24.785 | 0.040 | 0.764 | 0.00 | 0.00 | 0.00 | 0.00 |
| eil76 | 1.992 | 27.411 | 0.096 | 0.163 | 0.34 | 0.00 | 0.00 | 0.00 |
| pr76 | 0.816 | 27.793 | 1.228 | 0.039 | 1.12 | 0.40 | 0.00 | 0.00 |
| rat99 | 2.645 | 17.633 | 0.123 | 0.550 | 0.09 | 0.09 | 0.00 | 0.00 |
| kroA100 | 4.017 | 28.828 | 18.313 | 0.728 | 0.10 | 0.00 | 0.00 | 0.00 |
| kroB100 | 5.142 | 34.686 | 1.119 | 0.147 | 2.29 | 0.74 | 0.65 | 0.00 |
| kroC100 | 0.972 | 35.506 | 0.349 | 1.571 | 0.00 | 0.00 | 0.00 | 0.00 |
| kroD100 | 2.717 | 38.018 | 0.866 | 0.572 | 0.07 | 0.00 | 0.00 | 0.00 |
| kroE100 | 1.470 | 26.589 | 1.832 | 1.216 | 3.83 | 0.27 | 0.00 | 0.00 |
| rd100 | 3.407 | 30.432 | 1.725 | 0.003 | 0.08 | 0.00 | 0.00 | 0.00 |
| eil101 | 2.994 | 26.701 | 0.387 | 1.529 | 0.03 | 0.00 | 0.00 | 0.00 |
| lin105 | 1.739 | 34.902 | 1.867 | 0.606 | 0.00 | 0.00 | 0.00 | 0.54 |
| pr107 | 3.933 | 30.564 | 0.898 | 0.439 | 0.91 | 0.61 | 0.62 | 0.08 |
| pr124 | 3.677 | 70.146 | 10.322 | 0.755 | 1.02 | 0.60 | 0.08 | 0.58 |
| bier127 | 5.908 | 45.561 | 3.044 | 1.948 | 0.94 | 0.54 | 1.50 | 0.66 |
| ch130 | 3.182 | 39.090 | 0.709 | 3.519 | 0.29 | 0.06 | 0.00 | 0.00 |
| pr136 | 5.064 | 58.673 | 0.000 | 3.387 | 0.19 | 0.10 | 0.01 | 0.00 |
| pr144 | 7.641 | 55.837 | 1.526 | 3.581 | 0.80 | 0.50 | 0.39 | 0.00 |
| ch150 | 4.584 | 49.743 | 0.312 | 2.113 | 0.57 | 0.49 | 0.00 | 0.00 |
| kroA150 | 3.784 | 45.411 | 0.724 | 2.984 | 0.34 | 0.14 | 0.00 | 0.00 |
| kroB150 | 2.437 | 36.745 | 0.886 | 3.258 | 0.30 | 0.00 | 0.07 | 0.02 |
| pr152 | 7.494 | 33.925 | 0.029 | 3.119 | 1.69 | 0.83 | 1.19 | 0.68 |
| u159 | 7.551 | 38.338 | 0.054 | 1.020 | 0.82 | 0.00 | 0.00 | 0.00 |
| rat195 | 6.839 | 24.968 | 0.743 | 1.666 | 1.48 | 1.27 | 0.79 | 0.11 |
| d198 | 373.020 | 62.351 | 0.522 | 4.772 | 3.32 | 1.97 | 0.86 | 0.00 |
| kroA200 | 7.106 | 40.885 | 1.441 | 2.029 | 2.28 | 0.57 | 0.49 | 0.00 |
| kroB200 | 8.541 | 43.643 | 2.064 | 2.589 | 2.35 | 0.92 | 2.50 | 0.00 |
| **Mean** | **16.767** | **40.025** | **1.725** | **1.529** | **0.97** | **0.35** | **0.28** | **0.09** |

## B.3 EVALUATION ON REAL-WORLD INSTANCES

We evaluate our model on real-world TSPLIB instances with 50–200 nodes (Reinelt, 1991). The model is trained on TSP-100 dataset, and compared against state-of-the-art baselines including DIFUSCO (Sun & Yang, 2023), T2T (Li et al., 2023b), and Fast T2T (Li et al., 2024). The hyperparameter configurations for the diffusion-based baselines are as follows: DIFUSCO with $T_s = 50$; T2T with $T_s = 50$ and $T_g = 30$; Fast T2T (with guided sampling) with $T_s = 10$ and $T_g = 10$, and NExCO with $D_s = 20$ All diffusion-based methods are evaluated under the same settings, using greedy decoding followed by 2-Opt local search as post-processing. For consistency, the coordinates of each TSPLIB instance are normalized to the range $[0, 1]$.

## B.4 ABLATION STUDY ON THE CORRUPTION SCHEME

We compare our mask-based forward corruption with a uniform corruption scheme that treats all variables as equally likely to be flipped, similar to the uniform perturbation used in Fast-T2T (Ta-

ble 10). Across TSP-500 and TSP-1000, and under the same number of expansion steps, the mask-based model consistently achieves lower optimality gaps. The difference arises from the structural behavior of the two corruption processes. Uniform corruption injects noise indiscriminately and often produces partial states with weak or noisy supervision signals for the denoiser. In contrast, the proposed mask corruption applies one-way absorbing $1 \rightarrow 0$ updates that preserve the feasible structure of partial solutions while selectively revealing informative variables. This yields clearer denoising targets and a more stable reverse trajectory, which in turn explains the consistent improvements observed in our ablations.

Table 10: Performance comparison across different corruption scheme.

| Exp. Step | Scheme | TSP-500 | | | TSP-1000 | | |
|---|---|---|---|---|---|---|---|
| | | Obj. | Gap↓ | Time | Obj. | Gap↓ | Time |
| 3 | Uniform | 16.66 | 0.65% | 0.27s | 23.34 | 0.94% | 1.00s |
| | Mask | 16.61 | 0.39% | 0.23s | 23.31 | 0.85% | 0.91s |
| 5 | Uniform | 16.65 | 0.59% | 0.39s | 23.32 | 0.86% | 1.44s |
| | Mask | 16.59 | 0.28% | 0.33s | 23.26 | 0.63% | 1.31s |
| 7 | Uniform | 16.63 | 0.53% | 0.49s | 23.30 | 0.78% | 1.85s |
| | Mask | 16.59 | 0.25% | 0.43s | 23.24 | 0.52% | 1.68s |

### B.5 ABLATION STUDY ON THE ADAPTIVE EXPANSION

Table 11 reports the comparison between NAE and a non-adaptive "global t-schedule" baseline. The global baseline does not perform any form of adaptive expansion. It runs the diffusion model for a fixed number of denoising steps, produces a dense full prediction at the final step, and then applies a single greedy decoding to obtain a complete solution. No intermediate partial-state construction is carried out, and the amount of expansion is fixed rather than guided by model confidence. In contrast, NAE operates directly on feasible partial states and expands them progressively. At each step, it activates candidates according to their confidence scores and applies feasibility projection to maintain monotone growth of the partial solution. This native adaptivity enables the model to commit early to high-confidence regions while deferring uncertain components to later steps, thereby structuring the decoding trajectory in a principled way. As shown in the ablation, this constructive and confidence-aware expansion yields significantly smaller optimality gaps than the global non-adaptive schedule under the same number of denoiser calls. The improvement therefore stems not from additional computation, but from the design of an adaptive partial-state expansion mechanism.

Table 11: Comparison between Global and NAE under different expansion steps.

| Exp. Step | Method | TSP-500 | | | TSP-1000 | | |
|---|---|---|---|---|---|---|---|
| | | Obj. | Gap↓ | Time (s) | Obj. | Gap↓ | Time (s) |
| 3 | Global | 16.67 | 0.73% | 0.22s | 23.39 | 1.16% | 0.81s |
| | NAE | 16.61 | 0.39% | 0.23s | 23.31 | 0.85% | 0.91s |
| 5 | Global | 16.67 | 0.71% | 0.29s | 23.34 | 0.96% | 1.04 |
| | NAE | 16.59 | 0.28% | 0.33s | 23.26 | 0.63% | 1.31s |
| 7 | Global | 16.65 | 0.63% | 0.36s | 23.33 | 0.92% | 1.29s |
| | NAE | 16.59 | 0.25% | 0.43s | 23.24 | 0.52% | 1.68s |

### B.6 ABLATION STUDY ON THE FEASIBILITY PROJECTION

Table 12 compares our feasibility projection mechanism with a greedy decoding baseline. In the greedy baseline, a complete solution is constructed at every diffusion step by selecting variables according to their predicted probabilities. After obtaining this full solution, a fixed proportion of variables is remasked, and the resulting state is used as input to the next diffusion step. This produces a wrapper-like refinement cycle that repeatedly rebuilds full solutions throughout the trajectory.

In contrast, our projection operator $\Gamma$ maintains a single monotone partial-solution trajectory. At each step, $\Gamma$ accepts a candidate activation only when feasibility is preserved, and it does not generate full solutions prematurely. This prevents the repeated reconstruction inherent to greedy decoding and avoids the error accumulation introduced by successive remasking cycles. Empirically, both approaches can eventually achieve similar optimality gaps when sufficient steps are allowed. However, greedy decoding consistently incurs higher runtime and exhibits less stable behavior due to its reconstruct–remask procedure. These results demonstrate that the feasibility projection used by NAE offers a more efficient and principled alternative to wrapper-style greedy refinement strategies.

Table 12: Comparison between Greedy decoding and Projection under different expansion steps.

| Exp. Step | Method | TSP-500 | | | TSP-1000 | | |
| --- | --- | --- | --- | --- | --- | --- | --- |
| | | Obj. | Gap↓ | Time (s) | Obj. | Gap↓ | Time (s) |
| 3 | Greedy decoding | 16.61 | 0.39% | 0.39s | 23.33 | 0.94% | 1.29s |
| | Projection | 16.61 | 0.39% | 0.23s | 23.31 | 0.85% | 0.91s |
| 5 | Greedy decoding | 16.60 | 0.31% | 0.54s | 23.26 | 0.63% | 1.95s |
| | Projection | 16.59 | 0.28% | 0.33s | 23.26 | 0.63% | 1.31s |
| 7 | Greedy decoding | 16.59 | 0.26% | 0.71s | 23.25 | 0.57% | 2.59s |
| | Projection | 16.59 | 0.25% | 0.43s | 23.24 | 0.52% | 1.68s |

## C  MODEL ARCHITECTURE DETAILS

### C.1  INPUT EMBEDDING LAYER

Given node vector $x \in \mathbb{R}^{N \times 2}$, weighted edge vector $e \in \mathbb{R}^E$, denoising timestep $t \in \{\tau_1, \ldots, \tau_M\}$, where $N$ denotes the number of nodes in the graph, and $E$ denotes the number of edges, we compute the sinusoidal features of each input element respectively:

$$\tilde{x}_i = \text{concat}(\tilde{x}_{i,0}, \tilde{x}_{i,1}), \tag{9}$$

$$\tilde{x}_{i,j} = \text{concat}\left(\sin \frac{x_{i,j}}{T^{0/d}}, \cos \frac{x_{i,j}}{T^{0/d}}, \sin \frac{x_{i,j}}{T^{2/d}}, \cos \frac{x_{i,j}}{T^{2/d}}, \ldots, \sin \frac{x_{i,j}}{T^{d/d}}, \cos \frac{x_{i,j}}{T^{d/d}}\right), \tag{10}$$

$$\tilde{e}_i = \text{concat}\left(\sin \frac{e_i}{T^{0/d}}, \cos \frac{e_i}{T^{0/d}}, \sin \frac{e_i}{T^{2/d}}, \cos \frac{e_i}{T^{2/d}}, \ldots, \sin \frac{e_i}{T^{d/d}}, \cos \frac{e_i}{T^{d/d}}\right), \tag{11}$$

where $d$ is the embedding dimension, $T$ is a large number (usually selected as 10000), and $\text{concat}(\cdot)$ denotes concatenation. In CVRP, each node is described not only by its coordinates but also by customer demand $c_i$ and a depot indicator $\delta_i \in \{0, 1\}$. We embed these heterogeneous features separately and merge them into the node representation:

$$\tilde{c}_i = \text{concat}\left(\sin \frac{c_i}{T^{0/d}}, \cos \frac{c_i}{T^{0/d}}, \ldots, \sin \frac{c_i}{T^{d/d}}, \cos \frac{c_i}{T^{d/d}}\right), \tag{12}$$

$$\tilde{\delta}_i = \text{Embed}_{\text{depot}}(\delta_i), \tag{13}$$

$$\tilde{x}_i = \tilde{x}_i + \tilde{d}_i + \tilde{\delta}_i. \tag{14}$$

Next, we compute the input features of the graph convolution layer:

$$x_i^0 = W_1^0 \tilde{x}_i, \tag{15}$$

$$e_i^0 = W_2^0 \tilde{e}_i. \tag{16}$$

Specifically, for TSP and CVRP, the embedding input edge vector $e$ is a weighted adjacency matrix, which represents the distance between different nodes, and $e^0$ is computed as above. For MIS, we initialize $e^0$ to a zero matrix $0^{E \times d}$.

### C.2  GRAPH CONVOLUTION LAYER

Following (Joshi et al., 2019), the cross-layer convolution operation is formulated as:

$$x_i^{l+1} = x_i^l + \text{ReLU}(\text{BN}(W_1^l x_i^l + \sum_{j \sim i} \eta_{ij}^l \odot W_2^l x_j^l)), \qquad (17)$$

$$e_{ij}^{l+1} = e_{ij}^l + \text{ReLU}(\text{BN}(W_3^l e_{ij}^l + W_4^l x_i^l + W_5^l x_j^l)), \qquad (18)$$

$$\eta_{ij}^l = \frac{\sigma(e_{ij}^l)}{\sum_{j' \sim i} \sigma(e_{ij'}^l) + \epsilon}, \qquad (19)$$

where $x_i^l$ and $e_{ij}^l$ denote the node feature vector and edge feature vector at layer $l$, $W_1, \cdots, W_5 \in \mathbb{R}^{h \times h}$ denote the model weights, and $\eta_{ij}^l$ denotes the dense attention map. The convolution operation integrates the edge feature to accommodate the significance of edges in routing problems.

For TSP and CVRP, we aggregate the edge convolutional feature and reformulate the update for edge features as follows:

$$e_{ij}^{l+1} = e_{ij}^l + \text{ReLU}(\text{BN}(W_3^l e_{ij}^l + W_4^l x_i^l + W_5^l x_j^l)) + W_6^l(\text{ReLU}(t^0)). \qquad (20)$$

For MIS, we aggregate the node convolutional feature and reformulate the update for node features as follows:

$$x_i^{l+1} = x_i^l + \text{ReLU}(\text{BN}(W_1^l x_i^l + \sum_{j \sim i} \eta_{ij}^l \odot W_2^l x_j^l)) + W_6^l(\text{ReLU}(t^0)). \qquad (21)$$

### C.3 OUTPUT LAYER

The prediction of the edge heatmap in TSP and CVRP, and node heatmap in MIS is as follows:

$$e_{i,j} = \text{Softmax}(\text{norm}(\text{ReLU}(W_e e_{i,j}^L))), \qquad (22)$$

$$x_i = \text{Softmax}(\text{norm}(\text{ReLU}(W_n x_i^L))), \qquad (23)$$

where $L$ is the number of GCN layers and norm is layer normalization.

### C.4 MODEL PARAMETERS

For all tasks, we adopt a 12-layer GCN as described above. For TSP, following the setting of (Sun & Yang, 2023), we apply a K-Nearest Neighbor (KNN) strategy to sparsify the graph in order to reduce training memory and shrink the search space. Specifically, for TSP-500 and TSP-1000, the sparsity factors are set to 50 and 100, respectively.

## D EXPERIMENTAL SETUP

### D.1 HARDWARE

All models are trained and tested using NVIDIA A40 (48G) GPUs and Intel(R) Xeon(R) Gold 5220 CPU @ 2.20GHz. All test evaluations are performed in a single-threaded setting, where the average runtime per instance is reported to ensure fair comparison across different models.

### D.2 TRAINING SETUP

We have organized the training settings and model parameters of NExCO in Table 13. For all problems, we adopt a curriculum learning strategy, where models are progressively fine-tuned from smaller datasets to large ones.

## E LICENSES

The licenses for the codes used in this work are listed in Table 14.

Table 13: Details about the training hyperparameters of NExCO.

| Problem | Data | Data Size | Batch Size | Epoch | Learning Rate | Hidden Dimension |
|---------|------|-----------|------------|-------|---------------|------------------|
| TSP | Uniform-100 | 1,502k | 16 | 100 | 2e-4 | 256 |
| TSP | Uniform-500 | 128k | 6 | 50 | 2e-4 | 256 |
| TSP | Uniform-1000 | 64k | 4 | 50 | 2e-4 | 256 |
| MIS | RB-200-300 | 90k | 4 | 50 | 2e-4 | 256 |
| MIS | RB-800-1200 | 6.4k | 1 | 10 | 5e-5 | 256 |
| MIS | ER-700-800 | 163k | 4 | 50 | 2e-4 | 128 |
| CVRP | Uniform-50 | 1,280k | 32 | 50 | 2e-4 | 256 |
| CVRP | Uniform-100 | 640k | 12 | 50 | 2e-4 | 256 |
| CVRP | Uniform-200 | 32k | 2 | 50 | 2e-4 | 256 |

Table 14: Licenses for codes used in this work

| Resource | Type | Link | License |
|----------|------|------|---------|
| LKH3 (Helsgaun, 2017) | Code | http://webhotel4.ruc.dk/~keld/research/LKH-3/ | Available for academic research |
| HGS (Vidal et al., 2012)s | Code | https://github.com/chkwon/PyHygese | MIT License |
| Concorde (Applegate et al., 2006) | Code | https://github.com/jvkersch/pyconcorde | BSD 3-Clause License |
| KaMIS (Lamm et al., 2016) | Code | https://github.com/KarlsruheMIS/KaMIS | MIT License |
| AM (Kool et al., 2019) | Code | https://github.com/wouterkool/attention-learn-to-route | MIT License |
| BQ-NCO (Drakulic et al., 2023) | Code | https://github.com/naver/bq-nco | CC BY-NC-SA 4.0 |
| GCN (Joshi et al., 2019) | Code | https://github.com/chaitjo/graph-convnet-tsp | MIT License |
| DIMES (Qiu et al., 2022) | Code | https://github.com/DIMESTeam/DIMES | MIT License |
| DIFUSCO (Sun & Yang, 2023) | Code | https://github.com/Edward-Sun/DIFUSCO | MIT License |
| T2T (Li et al., 2023b) | Code | https://github.com/Thinklab-SJTU/T2TCO | MIT License |
| Fast T2T (Li et al., 2024) | Code | https://github.com/Thinklab-SJTU/Fast-T2T | MIT License |
| COExpander (Ma et al., 2025) | Code | https://github.com/Thinklab-SJTU/COExpander | Not specified |
| LEHD (Luo et al., 2023) | Code | https://github.com/CIAM-Group/NCO_code/tree/main/single_objective/LEHD | MIT License |
| Sym-NCO (Kim et al., 2022) | Code | https://github.com/alstn12088/Sym-NCO | Not specified |

## F LLM USAGE

In this work, LLMs were used solely to aid in the polishing and refinement of the manuscript text, including grammar correction, clarity improvement, and style consistency. All technical content, experimental results, and conclusions are the sole responsibility of the authors.

