# OpenReview forum: "NExCO: Native Solution Expansion for Diffusion-based Combinatorial Optimization"
_ICLR.cc/2026/Conference — ICLR 2026 Poster_

### Official Review · Reviewer_VX9M · 2025-10-29

**Soundness:** 4
**Presentation:** 4
**Contribution:** 4
**Rating:** 10
**Confidence:** 3

**Summary:**

Existing neural CO solvers either ensure local feasibility but lack global awareness (LC) or produce global predictions with constraint violations (GP). Current adaptive expansion is only an external wrapper with limited effectiveness.
NEXCO makes adaptive expansion native through CO-specific masked diffusion where intermediate states are meaningful partial solutions, combined with time-agnostic training and confidence-based progressive unmasking with feasibility projection.
The framework achieves about 50% quality improvement and 2~4 times speedup over state-of-the-art, successfully embedding adaptive expansion as an intrinsic generative principle rather than external wrapper.
NEXCO successfully realizes adaptive solution expansion as a native generative principle within masked diffusion, achieving superior performance across multiple CO problems. The framework opens new opportunities for integrating constructive expansion mechanisms into diffusion-based generative modeling, providing a step toward scalable and general-purpose neural solvers for combinatorial optimization.

**Strengths:**

1. Fundamental Reconceptualization

This is a strength because the shift from external wrapper to native integration addresses a fundamental architectural limitation. When adaptive expansion is just a wrapper (like COExpander), it must repeatedly call the underlying model, creating computational overhead and being limited by the backbone's capabilities. By making it native, NEXCO achieves the same adaptive behavior through the natural progression of the diffusion process itself, eliminating redundant computations and enabling tighter integration between the expansion logic and the generative model.

2. Theoretically Motivated Design

The CO-specific corruption is theoretically sound because it respects the sparse nature of CO solutions. Most CO problems have sparse solutions, so treating 0s and 1s asymmetrically aligns with the problem structure. This prevents the "conservative bias" problem shown in Figure 2, where symmetric masking causes models to prefer safe 0 predictions. The theoretical motivation directly translates to empirical success.

3. Computational Efficiency

The O(Ts) vs O(Ds Ts) complexity improvement is significant because it makes the method practical for real-world deployment.

**Weaknesses:**

1. Scalability Questions
The jump from 10K to 100K+ nodes represents a 10x increase that often reveals new bottlenecks. Memory requirements for storing full adjacency matrices grow quadratically, and the GNN message passing might become prohibitively expensive. The paper's use of KNN sparsification helps but doesn't fully address whether the core approach scales to industrial-size problems like million-node routing networks.

2. Training Data Requirements
NEXCO requires high-quality reference solutions for training (from Concorde for TSP, KaMIS for MIS). For new problem types where optimal solvers don't exist or are too slow, obtaining training data becomes a bottleneck. While Table 3 shows robustness to suboptimal labels, the method still needs some reasonable baseline solutions, limiting applicability to well-studied problems where such data exists.

**Questions:**

Please refer to the weaknesses.

---

> ### Author Response · Authors · 2025-11-19
>
> Dear Reviewer VX9M,
>
> Thanks for your insightful comments and for acknowledging our work. Below we respond to the specific comments.
>
> >**Q1: Scalability Questions The jump from 10K to 100K+ nodes represents a 10x increase that often reveals new bottlenecks. Memory requirements for storing full adjacency matrices grow quadratically, and the GNN message passing might become prohibitively expensive. The paper's use of KNN sparsification helps but doesn't fully address whether the core approach scales to industrial-size problems like million-node routing networks.**
>
> We appreciate the reviewer’s observation. Our current experiments focus on the 500–10k range, which is already beyond most prior NCO solvers; however, we agree that scaling to 100k+ or million-node instances requires additional system-level support. This limitation is primarily due to backbone message passing and memory usage rather than the NAE mechanism itself. To extend to larger regimes, NEXCO is naturally compatible with standard scaling strategies used in large-graph CO systems, such as divide-and-conquer frameworks (e.g., region partitioning similar to GLOP [1]), or replacing the backbone GNN with lightweight sparse-attention Transformers [2]. These directions do not alter the proposed generative principle and can be incorporated in future work to achieve industrial-scale deployments. We will clarify this limitation and discuss such practical extensions in the revised version (Appendix C.2).
>
> [1] GLOP: Learning Global Partition and Local Construction for Solving Large-scale Routing Problems in Real-time, AAAI, 2024.
>
> [2] Boosting neural combinatorial optimization for large-scale vehicle routing problems. ICLR 2025.
>
> >**Q2: Training Data Requirements NEXCO requires high-quality reference solutions for training (from Concorde for TSP, KaMIS for MIS). For new problem types where optimal solvers don't exist or are too slow, obtaining training data becomes a bottleneck. While Table 3 shows robustness to suboptimal labels, the method still needs some reasonable baseline solutions, limiting applicability to well-studied problems where such data exists.**
>
> We agree that obtaining optimal labels can be costly for new CO tasks. Fortunately, NEXCO does not require optimal supervision: Table 3 already shows that the model remains strong when trained on suboptimal or noisy solutions, suggesting that “optimality” is not essential as long as solutions are feasible. For many practical CO problems, high-quality heuristic solvers can produce such feasible labels at negligible cost, and these readily substitute for optimal ones. Moreover, the partial-solution semantics of NEXCO provide a natural mechanism for overcoming the scarcity and high cost of supervision in large-scale CO tasks. Because the model operates on feasible partial states rather than requiring complete high-quality solutions at every step, it can be trained using incomplete, heuristic, or low-cost labels and gradually refine its own predictions through iterative self-training. This significantly reduces reliance on exact solvers, whose computational cost grows prohibitively with instance size. The ability to learn from weak or partial supervision aligns with recent scalable training pipelines[1,2] that couple lightweight backbones with progressive bootstrapping. Incorporating such strategies offers a promising path for applying NEXCO in settings where high-quality ground-truth labels are expensive or unavailable, thereby further extending its applicability to truly large and industrial-scale CO environments.We will clarify this limitation and discuss such practical extensions in the revised version (Appendix C.2).
>
> [1] Boosting neural combinatorial optimization for large-scale vehicle routing problems. ICLR 2025.
>
> [2] Generation as Search Operator for Test-Time Scaling of Diffusion-based Combinatorial Optimization. NeurIPS 2025.

---

> > ### Comment · Reviewer_VX9M · 2025-11-22
> >
> > Thank you for the detailed explanation.
> > I now understand the things I was curious about.

---

### Official Review · Reviewer_Cjz9 · 2025-10-30

**Soundness:** 3
**Presentation:** 3
**Contribution:** 2
**Rating:** 4
**Confidence:** 2

**Summary:**

The paper introduces NEXCO, a masked-diffusion framework for neural CO that
(i) replaces uniform bit-flip noise with a CO-specific, one-way masking that only turns selected 1’s to 0 (never adding false positives),
(ii) trains a time-agnostic denoiser with time-agnostic optimization consistency (TOC), and
(iii) decodes via Native Adaptive Expansion (NAE)—progressively unmasking variables while a problem-specific projector enforces feasibility.
Claimed benefits: feasible partial states along the forward trajectory, schedule-free training, and constructive, efficient decoding; experiments are shown for TSP/MIS/CVRP with strong results.

**Strengths:**

The main strength and also the main novelty of this paper is that it proposes a masking diffusion process for combinatorial optimization, together with a constructive decoding process with feasibility projection, which addresses the “global heatmap decode” weakness in prior diffusion solvers.
The paper is well written and easy to follow.

**Weaknesses:**

The major weakness lies in the effectiveness of the algorithm proposed. Even though the masking-then-expansion framework is novel compared to the existing literature, it seems that this paper missed a very important related work [1] which also addresses the “global heatmap decode” weakness much earlier diffusion solvers like T2T and DIFUSCO. Instead of corrupting the partial solution in this paper, [1] corrupts the solution to its sub-optimal neighbors, which can also reduce the steps needed in reconstruction, and thus improves the speed of diffusion solver. I highly recommend authors to compare their work with this paper on benchmarks and analyze the pros/cons compared to this baseline, in addition to earlier diffusion solvers that have already been included.

[1] Generation as Search Operator for Test-Time Scaling of Diffusion-based Combinatorial Optimization (GenSCO)

**Questions:**

- Explicitly delineate what’s novel vs. "Generation as Search Operator for Test-Time Scaling of Diffusion-based Combinatorial Optimization (GenSCO)"

- Provide analysis theoretically / intuitively or empirically on the comparison of the proposed approach vs GenSCO

- Further ablation study to understand the source of gains and the importance of each component.

---

> ### Author Response · Authors · 2025-11-19
>
> Dear Reviewer Cjz9,
>
> Thank you for the detailed evaluation and constructive comments. Before addressing your specific questions, we would like to clarify the timeline regarding GenSCO. **Our ICLR submission was finalized and uploaded by the September 24 deadline, while the GenSCO preprint (later published at NeurIPS’25) was released publicly approximately four weeks afterward.** Consequently, GenSCO was not available for citation or comparison at submission time. We appreciate the reviewer for drawing our attention to this closely related work, and we provide a dedicated comparison and discussion below.
>
> >**Q1: Explicitly delineate what’s novel vs. "Generation as Search Operator for Test-Time Scaling of Diffusion-based Combinatorial Optimization (GenSCO)"**
>
> GenSCO treats generation as an external search operator: each cycle first disrupts a full solution via local search moves (e.g., 2-opt, bit flips), then uses a rectified flow / diffusion model to refine this disrupted solution, and repeats such cycles in an outer search loop. In contrast, our Native Adaptive Expansion (NAE) changes the internal structure of the diffusion decoder itself: intermediate states are always feasible partial solutions, and the model constructs the final solution through monotone expansion with feasibility projection, without an outer search loop.
>
> Concretely, the main conceptual differences are:
>
> **1.	refining solution vs. constructing solution**
>
> - GenSCO: Operates in a solution-to-solution regime: it learns rectified flows between suboptimal and (near-)optimal full solutions, and alternates disruption → refinement cycles in an outer search loop.
> - NAE (ours): Operates in a partial-solution regime: the forward corruption is one-way absorbing, so every diffusion state is a feasible partial solution, and the reverse process is a constructive expansion trajectory that never leaves the feasible region. This “partial-state view” and its finite-time convergence guarantee are not present in GenSCO.
>
> **2.	wrapper paradigm vs. native paradigm**
>
> - GenSCO: Adaptivity is controlled externally, via the number of search cycles, the disruption strength (e.g., number of 2-opt moves, flip ratio), and optional post-processing; the diffusion / flow model itself is used as a search operator inside this loop.
>
> - NAE: Adaptivity is native to the decoder: each diffusion step combines a confidence-ranked mask with feasibility projection, and the amount of expansion at each step is directly controlled by the learned heatmap and α, without any extra outer loop or hand-crafted disruption operators.
>
> **3.	Feasibility handling and decoding paradigm.**
>
> - GenSCO: Works with relaxed solutions in $[0,1]^N$, then uses greedy decoding plus optional local search (e.g., 2-opt) as post-processing after each operator cycle to recover feasibility.
>
>  - NAE: Enforces feasibility throughout the diffusion trajectory: our $\Gamma$ projector guarantees that every intermediate state remains a feasible partial solution, and we prove that NAE converges to a complete feasible solution in at most $N_{\max}$ steps under mild assumptions. This “always-feasible, monotone expansion” view is unique to our framework.
>
> In summary, GenSCO and NAE address different levels of the pipeline: GenSCO is an **scalable inference-time search wrapper** for diffusion-based solvers, whereas NAE proposes a new **native decoding principle** where intermediate states are feasible partial solutions with guaranteed finite-time expansion to a full solution.
>
> >**Q2: Provide analysis theoretically / intuitively or empirically on the comparison of the proposed approach vs GenSCO**
>
> NAE provides a finite-time convergence guarantee for its constructive expansion under mild assumptions, thanks to the one-way absorbing forward process and the monotone $\Gamma$ projector. GenSCO instead analyzes how rectified flows and search cycles improve scaling behavior at test time, but does not impose feasibility on all intermediate states. **The two perspectives are thus complementary: NAE focuses on structuring the diffusion path itself, while GenSCO focuses on how to reuse a diffusion/flow model as a search primitive.**
>
> Conceptually, GenSCO is well-suited for long-horizon search with many cycles, where its disruption–refinement loop can keep exploiting a strong base model. NAE is designed for one-pass adaptive decoding: it aims to extract as much quality as possible from a single constructive trajectory with limited denoiser calls. **In principle, NAE could serve as the underlying diffusion decoder inside a GenSCO-style search framework, indicating that the two are orthogonal and potentially combinable rather than overlapping.**

---

> ### Author Response · Authors · 2025-11-19
> **Official Comment by Authors (Cont.)**
>
> >**Q3: Further ablation study to understand the source of gains and the importance of each component.**
>
> Thank you for the helpful suggestion. The full experimental tables will be included in Appendix C.4–C.6 of the revised manuscript.
>
> **1. Ablation study on the corruption scheme**
>
> We compare our mask-based forward corruption with a uniform corruption scheme that treats all variables as equally likely to be flipped, similar to the uniform perturbation used in Fast-T2T. Across TSP-500 and TSP-1000, and under the same number of expansion steps, the mask-based model consistently achieves lower optimality gaps. The difference arises from the structural behavior of the two corruption processes. Uniform corruption injects noise indiscriminately and often produces partial states with weak or noisy supervision signals for the denoiser. In contrast, the proposed mask corruption applies one-way absorbing $1 \rightarrow 0$ updates that preserve the feasible structure of partial solutions while selectively revealing informative variables. This yields clearer denoising targets and a more stable reverse trajectory, which in turn explains the consistent improvements observed in our ablations.
>
> |Expansion Step| Method  | TSP500 Obj. | TSP500 Gap | TSP500 Time |TSP1000 Obj. | TSP1000 Gap | TSP1000 Time |
> | ------------- |:-------------:|:-------------:|:-------------:|:-------------:|:-------------:|:-------------:|:-------------:|
> 3|Uniform|16.66|0.65|0.27|23.34|0.94|1.00|
> ||Mask | 16.61| 0.39|0.23|23.31|0.85|0.91|
> 5| Uniform      | 16.65     | 0.59 |0.39 |23.32|0.86|1.44|
> || Mask      | 16.59     | 0.28|0.33|23.26|0.63|1.31|
> 7| Uniform     |16.63| 0.53    |0.49|23.30|0.78|1.85|
> ||Mask|16.59|0.25|0.43|23.24|0.52|1.68|
>
> **2. Ablation study on the adaptive expansion**
>
> We report the comparison between NAE and a non-adaptive “global t-schedule’’ baseline. The global baseline does not perform any form of adaptive expansion. It runs the diffusion model for a fixed number of denoising steps, produces a dense full prediction at the final step, and then applies a single greedy decoding to obtain a complete solution. No intermediate partial-state construction is carried out, and the amount of expansion is fixed rather than guided by model confidence. In contrast, NAE operates directly on feasible partial states and expands them progressively. At each step, it activates candidates according to their confidence scores and applies feasibility projection to maintain monotone growth of the partial solution. This native adaptivity enables the model to commit early to high-confidence regions while deferring uncertain components to later steps, thereby structuring the decoding trajectory in a principled way. As shown in the ablation, this constructive and confidence-aware expansion yields significantly smaller optimality gaps than the global non-adaptive schedule under the same number of denoiser calls. The improvement therefore stems not from additional computation, but from the design of an adaptive partial-state expansion mechanism.
>
>
> |Expansion Step| Method  | TSP500 Obj. | TSP500 Gap | TSP500 Time |TSP1000 Obj. | TSP1000 Gap | TSP1000 Time |
> | ------------- |:-------------:|:-------------:|:-------------:|:-------------:|:-------------:|:-------------:|:-------------:|
> 3|Global|16.67|0.73|0.22|23.39|1.16|0.81|
> ||NAE | 16.61| 0.39|0.23|23.31|0.85|0.91|
> 5| Global      | 16.67     | 0.71 |0.29 |23.34|0.96|1.04|
> || NAE      | 16.59     | 0.28|0.33|23.26|0.63|1.31|
> 7| Global     |16.65| 0.63    |0.36|23.33|0.92|1.29|
> ||NAE|16.59|0.25|0.43|23.24|0.52|1.68|

---

> ### Author Response · Authors · 2025-11-19
> **Official Comment by Authors (Cont.)**
>
> **3. Ablation study on the feasibility projection**
>
> We compare feasibility projection mechanism with a greedy decoding baseline. In the greedy baseline, a complete solution is constructed at every diffusion step by selecting variables according to their predicted probabilities. After obtaining this full solution, a fixed proportion of variables is remasked, and the resulting state is used as input to the next diffusion step. This produces a wrapper-like refinement cycle that repeatedly rebuilds full solutions throughout the trajectory.
>
> In contrast, our projection operator $\Gamma$ maintains a single monotone partial-solution trajectory. At each step, $\Gamma$ accepts a candidate activation only when feasibility is preserved, and it does not generate full solutions prematurely. This prevents the repeated reconstruction inherent to greedy decoding and avoids the error accumulation introduced by successive remasking cycles. Empirically, both approaches can eventually achieve similar optimality gaps when sufficient steps are allowed. However, greedy decoding consistently incurs higher runtime and exhibits less stable behavior due to its reconstruct–remask procedure. These results demonstrate that the feasibility projection used by NAE offers a more efficient and principled alternative to wrapper-style greedy refinement strategies.
>
>
> |Expansion Step| Method  | TSP500 Obj. | TSP500 Gap | TSP500 Time |TSP1000 Obj. | TSP1000 Gap | TSP1000 Time |
> | ------------- |:-------------:|:-------------:|:-------------:|:-------------:|:-------------:|:-------------:|:-------------:|
> 3|Greedy decoding|16.61|0.39|0.39|23.33|0.94|1.29|
> ||Projection | 16.61| 0.39|0.23|23.31|0.85|0.91|
> 5| Greedy decoding      | 16.60     | 0.31 |0.54 |23.26|0.63|1.95|
> || Projection      | 16.59     | 0.28|0.33|23.26|0.63|1.31|
> 7| Greedy decoding     |16.59| 0.26    |0.71|23.25|0.57|2.59|
> ||Projection|16.59|0.25|0.43|23.24|0.52|1.68|

---

> ### Author Response · Authors · 2025-11-27
> **Official Comment by Authors (Supplement to Q2)**
>
> Empirically, instantiating GenSCO’s iterative perturb–refine principle with our method is straightforward: in each cycle we use NAE (Expansion Step = 3) to produce a refined constructive solution and apply our stochastic masking to form the perturbed input for the next cycle, replacing GenSCO’s 2-opt–based disruptions. Because GenSCO’s test data is not publicly available, we report their official results (Table 10 in the original paper) together with our GCN-based NEXCO under matched cycle budgets. The results are shown below. They indicate that, when used within an iterative framework, NAE attains competitive or better solution quality with substantially lower runtime, consistent with the complementary roles of GenSCO’s long-horizon search and NAE’s efficient constructive decoding.
>
>
> TSP-500 Results
>
> |  **Method**               |  **Obj.**    |  **Gap**     |  **Time**   |
> | --------------------------- | --------------- | --------------- | -------------- |
> |  GenSCO (C=40, GCN)      |  16.556  |  0.060%  |  1m45s |
> |  Iterative NAE-(C=10)                |  16.555     |  0.042%      |  23.43 s     |
> |  Iterative NAE-(C=40)                   |  16.552     |  0.029%     |  1m31s     |
>
>
>
>
> TSP-1000 Results
>
> |  **Method**               |  **Obj.**    |  **Gap**     |  **Time**   |
> | --------------------------- | --------------- | --------------- | -------------- |
> |  GenSCO (C=40, GCN)  |  23.157  |  0.167% |  6m50s  |
> |  Iterative NAE-(C=10)                    |  23.143     |  0.111%      |  71.45 s     |
> |  Iterative NAE-(C=40)                  |  23.138     |  0.090%      |  4m32 s    |

---

### Official Review · Reviewer_PqJb · 2025-11-03

**Soundness:** 3
**Presentation:** 3
**Contribution:** 3
**Rating:** 6
**Confidence:** 4

**Summary:**

This paper introduces NEXCO, a diffusion-based framework for neural combinatorial optimization that makes adaptive expansion native to the generative model. The core idea is CO-specific masked diffusion that only drops active variables (1→0), a time‑agnostic GNN denoiser trained with optimization consistency across corruption levels, and an inference routine that expands solutions via confidence-ranked candidate sets with feasibility projection. Across TSP, MIS, and CVRP, NEXCO reports stronger solution quality and faster inference than prior LC/GP/AE baselines.

**Strengths:**

--Native AE design: Intermediate diffusion states are usable partial solutions, not just noisy heatmaps.
--Clear efficiency story: Explicit O(Ts) complexity and practical speedups (e.g., TSP‑500 gap 0.39% → 0.26% with ~2× faster; MIS ER drop 9.31% → 4.20% with >2× speedup; CVRP‑100 4.19% → 1.40%).
--Robustness and generalization: Cross‑scale transfer (Table 4) and training with suboptimal labels still yields strong solutions (Table 3).
--Breadth: Applies to TSP, MIS, and CVRP, including cases where GP diffusion struggles (CVRP).

**Weaknesses:**

--Theory is light: Convergence and approximation are argued intuitively; formal guarantees are not provided.
--Problem-specific components: Γ(·) requires tailored feasibility logic per task, which may limit plug‑and‑play generality.
--Sensitivity: Performance depends on expansion steps and candidate threshold α (Figures 3–4); guidance is empirical rather than principled.
--Scalability tactics: Large-scale TSP relies on KNN sparsification (§C.4, Table 8), which is effective but introduces an extra design choice.

**Questions:**

--Can you provide any formal convergence or approximation guarantees for NAE, even under simplified assumptions?
--How generalizable is Γ(·)? Could you outline patterns or templates to implement feasibility projection across new CO tasks?
--What is the runtime breakdown between denoiser calls and Γ(·) projection across tasks/scales?
--How robust is performance to α and Ds across distributions and instance scales? Any adaptive schemes that reduce tuning?

---

> ### Author Response · Authors · 2025-11-19
>
> Dear Reviewer PqJb,
>
> Thanks for your insightful comments and for acknowledging our work. Below we respond to the specific comments.
>
> >**Q1: Can you provide any formal convergence or approximation guarantees for NAE, even under simplified assumptions?**
>
> Thank you for raising this important point. In the revision (Appendix B.1), we have added a formal finite-time convergence guarantee for the Native Adaptive Expansion (NAE) procedure under mild and standard assumptions that hold for TSP, MIS, and CVRP.
> We provide a formal proposition of the Native Adaptive Expansion (NAE) procedure. Because the forward corruption in our CO-specific diffusion process is one-way absorbing ($1\rightarrow0$), every diffusion state remains a feasible partial solution. The reverse step expands this partial solution via $\mathbf{x}_{t+1} = \Gamma(\mathbf{x}_t \vee \mathbf{z}_t)$,where $\mathbf{z}_t$ is the candidate activation mask predicted by the denoiser, and $\Gamma$ is a feasibility projector. We assume the following mild and standard conditions, satisfied by the projectors used for TSP, MIS, and CVRP:
>
> **(A1) Monotone projection:**
> $\Gamma(\mathbf{x}) \succeq \mathbf{x} \quad \text{for all feasible } \mathbf{x}$.
>
> **(A2) Strict expandability:**
> $\exists\, \mathbf{z}_t \text{ such that }
> \Gamma(\mathbf{x}_t \vee \mathbf{z}_t) \succ \mathbf{x}_t
> \quad \text{whenever } \mathbf{x}_t \text{ is incomplete}$.
>
> **(A3) Bounded solution size:**
> $\text{Any complete feasible solution contains at most } N_{\max} \text{ active variables}$.
>
> **Proposition**: Under assumptions (A1)-(A3), NAE generates a monotone sequence
> $\textbf{x}_0 \preceq \mathbf{x}_1 \preceq \cdots$
> and converges to a complete feasible solution in at most $N_max$ iterations.
>
> The upper bound $N_{\max}$ is fully consistent with typical CO structures:
> - TSP: $N_{\max}=N$ edges in a Hamiltonian tour.
> - MIS: $N_{\max}\le n$ selected nodes.
> - CVRP: $N_{\max}$ equals the total number of edges across all valid routes.
>
> This analysis formalizes the intuition that NAE is a monotone constructive decoder that reaches a complete feasible solution in finite time, with complexity governed only by the number of denoiser calls.
>
> >**Q2: How generalizable is $\Gamma(\cdot)$? Could you outline patterns or templates to implement feasibility projection across new CO tasks?**
>
> We agree that CO tasks differ in their feasibility definitions, but the role of $\Gamma(\cdot)$ in NAE is intentionally generic and template-driven. To make this design principle explicit, we formalize the task-agnostic projection pattern that underlies all our experiments and revise the paragraph accordingly (Section 3.4).
>
> Across TSP, MIS, and CVRP, $\Gamma(\cdot)$ follows the same three-step template:
>
> S1.	Sort candidates by model confidence:
> $C = \text{Top-sorted}(p)$;
>
> S2.	Iterate through candidates (greedy insertion):
> For each candidate variable $i \in C$;
>
> S3.	Accept $i$ iff it satisfies a boolean feasibility predicate:
> \\[
> x'(i)=
> \begin{cases}
> 1, & \text{if } \mathrm{Feasible}(x,i)=\text{True}, \\\\
> 0, & \text{otherwise}.
> \end{cases}
> \\]
> where $\text{Feasible(·)}$ is a single boolean check, e.g.:
> - TSP: does adding edge $(u,v)$ keep degree $\le 2$ and avoid subtours?
> - MIS: is node $i$ nonadjacent to all active nodes?
> - CVRP: does adding edge keep residual capacity $\ge 0$?
>
> This gives the exact same operational structure across all problems:
> “sorted candidates → attempt insert → boolean feasibility check → accept/reject”.
> Thus the mechanism of $\Gamma$ is invariant; only the internal predicate differs. For a new CO task, implementing $\Gamma$ requires only define:
> $\text{Feasible}(x, i): \text{Does activating variable } i \text{ keep } x \text{ within } \Omega?$
> This is typically very simple (degree, adjacency, knapsack capacity, precedence, etc.).

---

> > ### Author Response · Authors · 2025-11-19
> > **Official Comment by Authors (Cont.)**
> >
> > >**Q3: What is the runtime breakdown between denoiser calls and Γ(·) projection across tasks/scales?**
> >
> > We have added detailed runtime profiling for all tasks and instance sizes. The updated results are summarized below:
> >
> > | Tasks          | Total Time (s) | Denoiser            | Γ(·) projection       |
> > |----------------|:--------------:|:-------------------:|:---------------------:|
> > | TSP100  | 0.018          | 0.016 (88.89%)      | 0.002 (11.11%)        |
> > | TSP500  | 0.097          | 0.081 (83.51%)      | 0.016 (16.49%)        |
> > | TSP1000 | 0.218          | 0.162 (74.31%)      | 0.056 (25.69%)        |
> > | MIS-RBsmall   | 0.033          | 0.026 (78.79%)      | 0.007 (21.21%)        |
> > | MIS-RBlarge   | 0.190          | 0.173 (91.05%)      | 0.017 (8.95%)         |
> > | MIS-ER         | 0.200          | 0.184 (92.00%)      | 0.016 (8.00%)         |
> > | CVRP50  | 0.022          | 0.017 (77.27%)      | 0.005 (22.73%)        |
> > | CVRP100 | 0.030          | 0.021 (70.00%)      | 0.009 (30.00%)        |
> > | CVRP200 | 0.078          | 0.052 (66.67%)      | 0.026 (33.33%)        |
> >
> > Across all settings, denoiser calls are the dominant cost, accounting for $66-92$% of total inference time. While the relative cost of $\Gamma(\cdot)$ naturally increases with instance size, because feasibility checks must be performed on a larger set of candidate variables, its absolute cost remains small in all cases (milliseconds). Most importantly, the overall complexity of NAE continues to be governed by $O(T_s)$ denoiser calls, and the projection step adds only a lightweight overhead that scales in a predictable manner with instance size.
> >
> >
> > >**Q4: How robust is performance to $\alpha$ and Ds across distributions and instance scales? Any adaptive schemes that reduce tuning?**
> >
> > Thank you for the insightful question. We summarize the robustness results and provide additional clarification on $\alpha$ and $D_s$.
> >
> > **1. $D_s$ yields a predictable and stable efficiency–quality trade-off.**
> >
> > As shown in Figure 3, increasing $D_s$ *consistently improves solution quality, and increases runtime nearly linearly,*
> > across all three problem families (TSP, MIS, CVRP) and all instance scales. This monotonic pattern follows directly from NAE’s constructive nature: more expansion steps allow more opportunities to activate high-confidence variables, and therefore does not require problem-specific tuning. Users simply choose $D_s$ based on their runtime/quality budget.
> >
> > **2. $\alpha$ has a wide “working interval”, consistent across scales and distributions.**
> >
> > Figure 4 shows a broad plateau where performance is nearly unchanged across $\alpha$ values, e.g., $[0.6,0.7]$ for TSP, $[0.1,0.3]$ for MIS, and $[0.5,0.7]$ for CVRP. This plateau appears in all tasks, remains stable across small/medium/large instances, and only degrades when $\alpha$ becomes extreme (overly aggressive vs. overly conservative), which is expected for any threshold-based selector. Thus, $\alpha$ behaves robustly within a stable interval rather than being sensitive to fine-grained tuning.
> >
> > **3. simpler adaptive scheme**
> >
> > In addition to the tuned $\alpha$ used in the main paper, we have explored a simple yet *principled adaptive scheme* derived from the target-size consistency of CO solutions. Each problem family has a known or reliably estimable solution cardinality (for example, TSP solutions contain exactly N active edges, MIS solutions exhibit stable size statistics across instances, and CVRP route lengths can be approximated from demand distributions). Based on this structural prior, we adapt $\alpha_t$ at each expansion step so that the number of variables exceeding the threshold matches the expected solution size：
> > $$
> > \|\\{i:p_t(i) > \alpha_t\\}\| \approx \text{target size}.
> > $$
> > This ensures that the activated candidate set remains size-consistent with the underlying CO problem, providing a principled way to determine $\alpha_t$ without manual tuning. Empirically, this rule produces behavior very similar to the tuned $\alpha$ used in our main experiments, while offering a lightweight, domain-grounded adaptive alternative.
> >
> > | TSP Size \Coefficient | 0.25 | 0.5  | 0.75 | **1**    | 1.25 | 1.5  |
> > |----------|------|------|------|------|------|------|
> > | 500      | 0.46 | 0.45 | 0.42 | **0.32** | 0.34 | 0.35 |
> > | 1000     | 1.05 | 0.96 | 0.73 | **0.69** | 0.78 | 0.92 |

---

### Author Response · Authors · 2025-11-30
**General Response**

Dear AC and Reviewers,

We sincerely thank the AC and reviewers for the time and effort in reviewing our paper. Overall, all reviewers `(PqJb, Cjz9, VX9M)` recognized the novelty, clarity, and empirical strength of our contribution (*Soundness: 3,3,4; Presentation:3,3,4;Contribution: 3,2,4. Rating:6,4,10*). We are encouraged that all three reviewers highlight the **core innovation** of making adaptive solution expansion native to diffusion models, transforming intermediate states into feasible partial solutions and enabling constructive, efficient, and schedule-free decoding.

Across reviews, NEXCO is acknowledged for its "fundamental architectural improvement" over LC/GP/AE paradigms `(PqJb, VX9M)`, its "theoretically motivated corruption and decoding design" `(VX9M)`, and its "strong empirical gains", about 50% quality improvement and 2–4× speedup across TSP, MIS, and CVRP `(PqJb, Cjz9, VX9M)`. We greatly appreciate these positive evaluations.

All concerns raised by reviewers have been thoroughly addressed in the revision:

- Theory `(PqJb)` . We added a **formal finite-time convergence guarantee for NAE**, establishing monotone constructive expansion under mild and standard assumptions (Section 3.4);
- Generality of the feasibility projector Γ(·) `(PqJb)` . We formalized a clear **task-agnostic template for implementing the feasibility projector Γ(·)**, showing that the mechanism is generic and only requires a boolean feasibility predicate (Section 3.4);
- Runtime breakdown `(PqJb)` . We provided a **runtime breakdown across all tasks and scales**, demonstrating that denoiser calls dominate the cost (70–92%) while Γ(·) remains lightweight even on large instances (Appendix B.7);
- Sensitivity to α and Ds `(PqJb)` .  Beyond the parameter studies in Figures 3–4, we expanded the **robustness analysis** for both α and Ds, and introduced a **principled adaptive scheme** based on target-size consistency across different CO tasks (Section 4.4).
- Relation to GenSCO [1] `(Cjz9)` . Although **GenSCO (NeurIPS`25) was publicly released after the ICLR submission deadline**, we clarified the fundamental conceptual differences between its **solution-to-solution perturb–refine (wrapper) search** and our **native constructive expansion within diffusion**. We further added theoretical, intuitive, and empirical contrasts, showing that the two paradigms have **complementary strengths**.
- Albation Study `(Cjz9)` .  We conducted a set of expanded **ablation studies** dissecting the contributions of **(i)** mask-based corruption vs. uniform corruption, **(ii)** adaptive expansion vs. global non-adaptive schedules, and **(iii)** feasibility projection vs. greedy reconstruction. These ablations reveal that the main gains stem from NAE’s constructive partial-state semantics and consistent feasibility handling (Appendix B.4–B.6).
- Scalability and training-data concerns `(VX9M)`. We clarified that scaling limitations come from the backbone message passing, not from NEXCO’s generative principle, and discussed compatibility with standard large-graph scaling strategies (divide-and-conquer, sparse attention). We also emphasized that NEXCO does not require optimal supervision and can effectively learn from suboptimal labels (Table 3), aligning with recent scalable weak-supervision pipelines (Appendix B.2). The reviewer `(VX9M)` responded positively to these clarifications and indicated that the explanation addressed the points they were concerned about.


---

[1] Generation as Search Operator for Test-Time Scaling of Diffusion-based Combinatorial Optimization (GenSCO), NeurIPS 2025.

---

Thank you again for your time, consideration, and constructive input.

Best regards,

The Authors

---

### Meta-Review · Area_Chair_V9hb · 2025-12-29

**Summary:**

This work proposes NEXCO, a Native Adaptive Expansion for Combinatorial Optimization method, for neural combinatorial optimization. The proposed NEXCO method carefully incorporates the masked diffusion model into the adaptive expansion (AE) paradigm, which makes AE a native intrinsic decoding step during the diffusion process. The key components of NEXCO are: 1) a CO-specific one-way corruption approach that only drops active entry (1) to background (0), which ensures all intermediate states correspond to valid partial solutions; 2) a time-agnostic graph denoiser; and 3) a native adaptive expansion decoding strategy for feasibility projection. Experimental studies show NEXCO can achieve promising performance on different MIS/TSP/CVRP instances.

The reviewers originally had mixed scores for this work (10, 6, 4) and raised concerns on theory, the generality of the feasibility projector, relation to a closely-related work (GenSCO), scalability, availability of training data, more ablation studies and analysis. After the rebuttal, most of these concerns have been properly addressed and I believe Reviewer Cjz9 will raise the score from 4 to 6. Therefore, a reasonable final score of this work could be (10, 6, 6).

I read the paper in detail, and agree with all reviewers that this paper is well-written and easy to follow, the proposed method is novel with fundamental improvement, and the obtained results are promising. I think NEXCO has the potential to be an elegant and practical approach for the adaptive expansion paradigm of NCO. Therefore, I recommend accepting this work.

**Reviewer Concerns:**

I believe most concerns raised by the reviewers have been properly addressed by the rebuttal. Especially, the major concern of the reviewer with the most negative score (Reviewer Cjz9 with score 4) is on the relation with the closely-related work GenSCO [1]. The authors have correctly pointed out that the GenSCO preprint (later published at NeurIPS 2025) was released publicly approximately four weeks after the ICLR submission deadline. In addition, a detailed discussion with GenSCO is also provided in the rebuttal.

A minor remaining concern might be the scalability of NEXCO. This paper claims that the poor scalability is the major shortcoming of the local construction (LC) approach for NCO, which NEXCO can overcome. However, in this work, NEXCO is only tested on small-scale TSP/CVRP instances with up to 1K nodes. To my understanding, some LC methods, such as [2], can already tackle TSP/CVRP instances with 100K nodes. A proper discussion/comparison and clarification could be needed in the revised paper.

[1] Generation as Search Operator for Test-Time Scaling of Diffusion-based Combinatorial Optimization (GenSCO), NeurIPS 2025.

[2] Boosting neural combinatorial optimization for large-scale vehicle routing problems. ICLR 2025.

**Reviewer Scores:**

If the reviewers had been able to participate fully in the discussion, I think Reviewer VX9M and Reviewer PqJb will keep their original positive score (10, 6) and Reviewer Cjz9 will raise their score from 4 to 6.

---

### Decision · Program_Chairs · 2026-01-26

Accept (Poster)